# RACE: REAL-TIME ADAPTIVE CAMERA-INTRINSICS ESTIMATION VIA CONTROL THEORY

## ABSTRACT

Modern embodied AI systems, from mobile robots to AR devices, rely on accurate camera intrinsics to ensure reliable perception. Yet in real-world operation, the intrinsics drift due to heating, zoom events, mechanical shocks, a single hard landing, or simply incorrect factory calibration, thereby violating the fixed-parameter assumption that underpins most vision and learning pipelines. This induces a distribution shift in the visual input, which in turn degrades the performance of downstream models and tasks that rely on stable camera geometry. We introduce RACE (Real-time Adaptive Camera-intrinsic Estimation), a provably stable online learning algorithm that continually estimates camera intrinsics directly from a continuous monocular image stream. RACE updates parameters through a Lyapunov-stable adaptive law, guaranteeing global asymptotic convergence of the reprojection error dynamics and recovery of the true intrinsics under persistent excitation. Unlike prior batch optimization, heuristic self-calibration, or learning-based approaches, RACE requires no training data, bundle adjustment, or retraining. It provides the first theoretical bridge between adaptive control and online learning for camera models. Empirically, we evaluated RACE across public benchmarks (EuRoC, TUM, and TartanAir), demonstrating that it matches or surpasses state-of-the-art learning-based calibration while adapting in real-time with negligible computational overhead. Our results highlight RACE as a new class of theoretically grounded continual learners for camera intrinsics, enabling robust long-term perception in embodied agents.

*Link to code (anonymized for review, will be made public later):*
`https://anonymous.4open.science/r/race_iclr2026-FEFF`

## 1 INTRODUCTION

Accurate camera intrinsics (focal length, principal point, distortion) are foundational to computer vision, robotics, Simultaneous Localization and Mapping (SLAM), 3D reconstruction, and Augmented Reality (AR). In deployment, however, intrinsics are *not* constant: heating, zoom operations, vibrations, and shocks induce gradual or abrupt drift that invalidates the 'fixed intrinsics' assumption of standard vision pipelines. Left uncorrected, such errors silently corrupt downstream estimates: depth maps distort, maps warp, and AR overlays misalign. In safety-critical domains such as aerial robotics or autonomous driving, even minor reprojection errors can cascade into catastrophic failures. Yet most systems still rely on a one-time laboratory calibration, assuming those parameters remain valid indefinitely. This gap motivates the need for *online intrinsic calibration algorithm*: that can *calibrate camera online from scratch and adapt continuously* during operation.

Conventional remedies leave a gap and fall short in a few ways. Target-based methods (e.g., planar checkerboards) achieve high precision but require pausing operation and controlled scenes (Zhang, 2000). Self-calibration via SfM or bundle adjustment can refine intrinsics without targets, but runs in batch and suffers from degeneracies under limited motion or weak texture. Modern pipelines such as COLMAP include intrinsics in BA but still operate offline (Schönberger & Frahm, 2016). Learning-based methods regress intrinsics from images or integrate differentiable BA into deep SLAM, improving automation but demanding large training data and heavy compute. Generalization to unseen cameras and scenes remains fragile as a network trained on one camera rig or environment frequently mispredicts when faced with a new lens or lighting conditions (Workman et al., 2015;

Bogdan et al., 2018; Tang & Tan, 2018; Hagemann et al., 2023; Teed & Deng, 2021). Online estimators in VIO/SLAM sometimes include intrinsics in the state, but face inconsistency/observability issues and typically lack formal stability guarantees for the intrinsic update itself (Nobre et al., 2017; Yan et al., 2023). In short, we lack a method that is simultaneously (i) truly online, (ii) training-free, (iii) provably stable, and (iv) real-time on commodity CPUs.

We introduce *RACE* (Real-time Adaptive Camera-intrinsic Estimation), which frames intrinsic *calibration* as an *adaptive control* problem. Treating intrinsics as dynamic states, RACE applies a lightweight Lyapunov-based update driven by reprojection errors. Under standard *persistent excitation* (PE), i.e., sufficiently rich motion and scene variation, we (1) prove global Lyapunov stability of the error dynamics; (2) establish asymptotic convergence in the noise-free case; and (3) guarantee global uniform ultimate boundedness (GUUB) under bounded noise. The analysis trivially extends to radial-distortion parameters. Practically, RACE performs intrinsic calibration online: it is training-free, requires no bundle adjustment, and runs in real-time on a single CPU core without the need for GPU acceleration.

Our contributions are fourfold:

- *Provable Online Stability and Convergence:* We develop a Lyapunov-based adaptive law that treats intrinsics as dynamic states. Under appropriate persistent excitation conditions, we prove global uniform boundedness of the estimation errors in the presence of bounded noise and asymptotic convergence in the noise-free case.

- *Lightweight Real-Time Performance:* RACE runs entirely on a single CPU core with a simple update law, without any need for offline bundle adjustment or GPUs, and adding only **8.53** ms of per-frame overhead.

- *High Accuracy Across Benchmarks:* On EuRoC, TUM RGB-D, and TartanAir, RACE achieves subpixel RMS reprojection error, matching or outperforming state-of-the-art batch and learning-based calibration methods. On TUM RGB-D and EuRoC, with monocular input, RACE establishes new state-of-the-art accuracy, reduces the minimum reprojection error by up to **94%** among methods with zero failures.

- *Generalization Without Training:* By relying solely on visual reprojection errors, RACE adapts seamlessly to distortion models and challenging environments, remaining robust to measurement noise, zoom shifts, thermal drift, and sudden perturbations, without retraining or scene-specific priors.

RACE bridges the longstanding divide between laboratory calibration practices and the demands of lifelong autonomy by providing a provably stable, real-time, and training-free solution. This work can catalyze a new research direction at the intersection of control theory and computer vision, inspiring future adaptive mechanisms that continuously safeguard the integrity of perception systems in ever-changing real-world environments.

## 2 RELATED WORK

### 2.1 TRADITIONAL CAMERA CALIBRATION

Camera calibration has a rich history in computer vision and photogrammetry Liao et al. (2023). Classical methods use known targets (checkerboards, grids) to establish 3D–2D point correspondences and solve for intrinsics via closed-form or non-linear optimization. The method introduced by Zhang (2000) employs a flexible technique that utilizes a planar pattern observed from unknown orientations, which has since been widely adopted. Similarly, toolbox frameworks like Kalibr calibrate cameras offline before deployment Furgale et al. (2013). These target-based procedures are inherently offline and can achieve high accuracy in controlled settings.

Another line of classical work is self-calibration (or auto-calibration) from unknown scenes, which eliminates the need for dedicated patterns. Observing multiple scene images, one can recover intrinsics using structure-from-motion (SfM) or multi-view constraints, as demonstrated in early studies Hagemann et al. (2023); Zhu et al. (2023). Modern SfM pipelines, such as COLMAP, refine intrinsics as part of bundle adjustment Schönberger & Frahm (2016), eliminating the need for special

targets. Given enough views, bundle adjustment frameworks can jointly optimize intrinsics alongside poses and 3D structure. However, these approaches still operate in batch mode on collected data and can suffer from degeneracies in certain motions or scenes.

## 2.2 DEEP-LEARNING BASED CALIBRATION

Deep learning has been applied to automate camera calibration from images He et al. (2025). Early regression-based models trained convolutional networks to directly predict intrinsics (focal length, distortion, etc.) from a single image. Workman et al. (2015) employed a CNN to estimate focal length from unconstrained photos. Subsequent methods were extended to wider camera models, as seen in Bogdan et al. (2018), introducing DeepCalib—a deep network that regresses focal length and distortion for wide field-of-view cameras. These supervised approaches eliminate the need for manual calibration targets but require large annotated datasets of images with known intrinsic parameters. In practice, these models do not generalize. A network may perform well on the specific cameras or data it was trained on, but struggle on others without retraining. Attempts to train a single model across many devices have an accuracy trade-off and demand diverse training data to cover all possible camera types. Recent works have explored self-supervised or hybrid learning strategies to enhance robustness and reduce reliance on labeled data. These methods incorporate geometric consistency losses, allowing networks to learn calibration from multiple views or videos without direct supervision. For instance, Fang et al. (2022) enforces multi-view constraints by minimizing reprojection errors or aligning feature tracks across frames. This approach often involves embedding a differentiable bundle adjustment or structure-from-motion module into the network, combining deep feature extraction with classical optimization. Tang & Tan (2018) propose BA-Net, which integrates bundle adjustment into a CNN. Hagemann et al. (2023) exemplifies this by combining an intrinsic optimization layer into a deep SLAM network (DROID-SLAM Teed & Deng (2021)), allowing the system to infer camera intrinsics from monocular video during inference. Such techniques can effectively self-calibrate a camera as the network processes video, yielding high accuracy on benchmarks. Despite these advances, deep learning-based solutions have apparent drawbacks. They involve large models, expensive offline training phases, and lack theoretical convergence guarantees. Performance can degrade if the camera motion or scene falls outside the training distribution since generalization to novel conditions or camera models is not ensured without extensive retraining. In summary, while learning-based calibrators automate the process, they sacrifice the reliability and guarantees needed for lifelong deployment.

## 2.3 ONLINE AND LONG-TERM CALIBRATION IN PRACTICE

The challenges of maintaining calibration during long-term operations have prompted research into online techniques. Recent work in robotics and autonomous vehicles acknowledges that calibration must be updated regularly as conditions change Yan et al. (2023). Some approaches augment SLAM or visual-inertial odometry (VIO) systems with self-calibration capabilities Du & Brady (1993); Huang et al. (2020). For example, filter- and smoothing-based methods can treat intrinsics as part of the state vector and continuously estimate them along with motion. An early difficulty with filtering intrinsics is inconsistency and unobservability under certain motions Nobre et al. (2017). In autonomous driving, researchers have demonstrated self-supervised online calibration Heng et al. (2013). Other work has tackled online multi-sensor calibration (e.g. camera-LiDAR alignment) using neural networks and continuous sensor data. These efforts demonstrate a growing consensus that calibration should be an ongoing process, rather than a one-time initialization step. However, current online methods often remain dependent on learning, with the associated training overhead and potential domain limits, or implement heuristic filtering solutions without formal stability guarantees.

## 2.4 COMPARISON TO RACE

Our approach, RACE (Real-time Adaptive Camera-intrinsic Estimation), directly addresses the above gaps. Unlike prior methods, it requires no pre-calibration, training data, or learned priors, and can operate from a completely uncalibrated camera. Instead of batch optimization or offline learning, RACE treats intrinsics as dynamic states and updates them continuously from visual feedback. This control-theoretic formulation yields three key advantages: **(i)** RACE comes with formal

guarantees. We prove that the online learning algorithm remains stable under standard PE conditions, and the intrinsic error converges asymptotically towards zero. To our knowledge, it is the first method to achieve continuous intrinsic calibration with proven global stability. **(ii)** As it does not rely on learned priors, RACE naturally generalizes across different environments and camera models. The algorithm can also use generic visual cues, making it broadly applicable without retraining or customization. **(iii)** RACE runs in real-time with a lightweight update law, adding only a few milliseconds of overhead per frame.

In summary, RACE bridges the gap between traditional batch self-calibration and lifelong autonomy by providing a provably stable, real-time solution that continuously adapts to changing conditions.

## 3 METHODOLOGY

We begin by introducing the pinhole camera model (Sec.3.1), and then cast intrinsic adaptation as an online regression problem (Sec.3.2). We establish stability and convergence guarantees for the pinhole camera model in Sec.3.3, extend the framework to lens distortion in Sec.3.3.2, and analyze robustness under bounded noise in Sec . 3.3.3.

### 3.1 PINHOLE CAMERA MODEL

We adopt the standard pinhole camera model Hartley & Zisserman (2003). Let the (unknown) ground-truth intrinsics be $\theta^\star = [\, f_x^\star, \ f_y^\star, \ c_x^\star, \ c_y^\star\,]^\top$, with focal lengths $(f_x^\star, f_y^\star)$ and principal point $(c_x^\star, c_y^\star)$. Let $\mathbf{X}_w = [X_w, Y_w, Z_w]^\top$ denotes the 3D points in world frame and $(R, \mathbf{t}) \in SO(3) \times \mathbb{R}^3$ as the camera pose. Points in the camera frame can be written as $X_c = R\mathbf{X}_w + \mathbf{t}$, where $X_c = [X, Y, Z]$. The ideal pixel reprojection with zero skew is $u_t^\star = f_x^\star \frac{X}{Z} + c_x^\star$, $v_t^\star = f_y^\star \frac{Y}{Z} + c_y^\star$, and we write $p_t^\star = (u_t^\star, v_t^\star)$. This reprojection can be rewritten as a compact *linear regression* form

$$p_t^\star = \Phi_t\,\theta^\star, \qquad \Phi_t = \begin{bmatrix} \dfrac{X}{Z} & 0 & 1 & 0 \\[2mm] 0 & \dfrac{Y}{Z} & 0 & 1 \end{bmatrix}, \text{ where } \Phi_t \in \mathbb{R}^{2\times 4} \text{ is the per-feature regressor}$$

encoding the sensitivity of the image coordinates to each intrinsic parameter.

### 3.2 PROBLEM FORMULATION

Our objective is to design an *online learning algorithm* that continually adapts a camera's intrinsic parameters from a monocular image stream. The learner must (i) drive reprojection error $\tilde{p} \to 0$, (ii) *provably* converge to the true intrinsics over time, and (iii) remain robust to both parameter drift and sensor noise. Reprojection errors occur when the camera is miscalibrated or its intrinsic parameters drift away from their true values. RACE addresses this by updating the intrinsic estimate $\hat{\theta}_t$ such that the intrinsic error $\tilde{\theta}_t = \hat{\theta}_t - \theta^\star$ converges to zero online.

**Assumptions:** Consistent with standard calibration methods, we assume access to 2D–3D correspondences and known camera pose. This assumption allows us to isolate and rigorously evaluate the convergence behavior of RACE. Convergence guarantees require the standard *persistent excitation* (PE) condition, that is, sufficient diversity in motion and feature diversity over time to excite all parameters. We analyze robustness to PE violations in Appendix. C.5.

**Error dynamics:** Stacking the $N_t$ correspondences observed at time $t$ yields the compact relation

$$\tilde{p}_t = \Phi_t\,\tilde{\theta}_t, \qquad \tilde{p}_t = \hat{p}_t - p_t^\star, \tag{1}$$

The regressor $\Phi_t \in \mathbb{R}^{2N_t \times 4}$ encodes the sensitivity of pixel coordinates to changes in the intrinsic parameters. Importantly, $\Phi_t$ is *directly measurable from data* at each timestep, enabling a fully online update law for $\hat{\theta}_t$.

### 3.3 STABILITY AND CONVERGENCE ANALYSIS

We now present the theoretical analysis of our online learning algorithm RACE, focusing on the stability and convergence of both the reprojection error ($\tilde{p}$) and the parameter error ($\tilde{\theta}$). Our anal-

ysis follows standard assumptions from adaptive control theory and treats calibration as an online learning problem with provable guarantees rather than a nonlinear optimization problem. Complete derivations and proofs are provided in Appendix A.

### 3.3.1 IDEAL PINHOLE, NOISE-FREE CASE

We begin with the ideal noise-free pinhole setting to illustrate the core stability properties of RACE. We demonstrate that the proposed adaptive update law is stable and ensures convergence, as reprojection errors vanish asymptotically, and under PE, the estimated intrinsics converge to their actual values.

For stability analysis, we define the positive-definite Lyapunov candidate (see Appendix B.1 for a primer on control theory):

$$V(t) = \frac{1}{2}\tilde{\theta}_t^\top \Gamma^{-1} \tilde{\theta}_t \tag{2}$$

where $\Gamma = \Gamma^\top \succ 0$ is the diagonal adaptation gain matrix, $\Gamma = \mathrm{diag}(\gamma_{fx}, \gamma_{fy}, \gamma_{cx}, \gamma_{cy})$. Differentiating $V(t)$ with respect to time and substituting the adaptive update law, $\dot{\hat{\theta}}_t = -\Gamma \Phi_t^\top \tilde{p}_t$, it can be shown that (see Appendix A.1) the time derivative simplifies to:

$$\dot{V}(t) = -\tilde{p}_t^\top \tilde{p}_t \leq 0. \tag{3}$$

It should be noted that $\dot{V}(t)$ is non-increasing, under the assumption that $\Phi(t)$ is bounded and all closed-loop signals are bounded for any fixed $\Gamma \succ 0$. The Barbalat's Lemma (Appendix B.1) guarantees convergence of parameters under the PE condition. Additionally, $\Gamma$ is tuned once and shows impressive results on diverse real-world benchmarks, see Appendix C.1. Based on these dynamics and the adaptive law, we can now establish formal stability and convergence guarantees.

> **Theorem 1 (Continuous-Time Stability & Convergence)** *Consider the adaptive intrinsic calibration system equation 1, under the assumption that $\Phi(t)$ is bounded. Then, all signals in the closed loop system are bounded, and the reprojection errors $\tilde{p}(t)$ asymptotically converge to zero. Furthermore, if the regressor is persistently exciting equation 14, then the parameter estimation error $\tilde{\theta}(t) \to 0$ as $t \to \infty$.*

Intuitively, Theorem 1 establishes that RACE behaves as a stable online learning algorithm. It guarantees driving reprojection error to zero for any arbitrary initialization, and under sufficient motion diversity (the PE condition), it also recovers the true intrinsic parameters. This property is essential for long-term deployment, since even minor calibration errors can accumulate and degrade downstream perception.

We empirically validate these guarantees in Sec. 4, demonstrating that RACE reduces reprojection error and recovers accurate intrinsics even under large initialization errors (Sec. 4.1, Appendix C.3). Furthermore, it remains robust and adaptive to long-duration drifts in parameters (Appendix C.7).

### 3.3.2 COMPENSATION FOR LENS DISTORTION

Real cameras rarely conform to the ideal pinhole model: radial and tangential distortion introduce nonlinear effects that must be compensated. To handle this, we can extend the estimator $\hat{\theta}$ with additional distortion parameters $\mathbf{d} = [k_1, k_2, p_1, p_2]^\top$, depending on task requirements. The augmented model captures distortion by making pixel coordinates nonlinear functions of the normalized image radius. We adopt the classical Brown–Conrady radial–tangential model Conrady (1919); full details are provided in Appendix B.3.

> **Remark 1** *The regressor $\Phi_t$ is adapted to this distortion-aware model, and the same update law is applied to the expanded parameter vector $\theta = [f_x, f_y, c_x, c_y, \mathbf{d}]^\top$. Convergence guarantees continue to hold under the boundedness and persistent excitation (PE) assumptions.*

We omit a formal proof, since the convergence analysis follows directly from Theorem 1. We validate the performance of the distortion-aware estimator in real-world datasets. Including a single radial distortion term does not increase the computational complexity of our estimator and can be

seamlessly integrated into the adaptive update law C.2. While adding additional distortion parameters may slow convergence, they do not compromise stability. In practice, distortion coefficients can be initialized to zero without adverse effects on convergence.

### 3.3.3 BOUNDED NOISE

In real deployments, image measurements are corrupted by sensor noise, quantization, motion blur, environmental factors, and correspondence errors. We model this as $u_{\text{obs}} = u + \eta_u$, $v_{\text{obs}} = v + \eta_v$, with $\eta_{u,v}$ zero-mean disturbances bounded by $\pm\eta_{\max}$. This yields perturbed reprojection error dynamics:

$$\tilde{p} = \Phi\,\tilde{\theta} + n, \qquad n = [\eta_u, \eta_v]^\top. \tag{4}$$

The adaptive update law remains unchanged, but the Lyapunov derivative acquires an additional disturbance term, $n_t$. Standard results on perturbed adaptive systems then imply *ultimate boundedness*: all signals remain bounded, and the reprojection error converges to a ball whose radius scales with $\eta_{\max}$.

> **Theorem 2 (Robustness under Bounded Measurement Noise)** *Consider the adaptive intrinsic calibration system equation 1, with additive measurement noise $n(t)$ such that $\|\tilde{p}_t\| \leq \|\Phi_t\tilde{\theta}_t\| + \|n_t\|$, $\|n_t\| \leq \eta_{max}$ for some $\eta > 0$. Assume that $\Phi_t$ and $\dot{\Phi}_t$ are bounded and $\Phi_t$ is persistently exciting. Then, under the adaptive update law, the parameter error $\tilde{\theta}_t$ is globally uniformly ultimately bounded (GUUB).*

Theorem 2 guarantees robustness, ensuring that bounded measurement noise does not destabilize the online learning estimator; the intrinsic error remains confined to a neighborhood whose size scales with the magnitude of the disturbance. In practice, this means that even under moderate degradation (e.g., low light or motion blur), RACE converges to parameters accurate enough for downstream perception tasks. As $\eta \to 0$, the result smoothly recovers the asymptotic convergence guarantee of Theorem 1.

We validate this robustness on diverse real-world sequences with controlled injected noise (see Sec. 4.1, Appendix C.4). Across all settings, the intrinsic error remains bounded within a tight envelope proportional to the disturbance level, confirming that RACE maintains stable, real-time calibration under practical non-idealities.

**In summary,** our theoretical analysis shows that the adaptive estimator is asymptotically convergent in the ideal noise-free case (Theorem 1) and remains globally uniformly ultimately bounded under realistic measurement noise (Theorem 2). The same framework also extends naturally to distortion-aware camera models (Remark 1). For completeness, we summarize RACE as pseudo code in Algorithm 1 (Appendix D.1). We now turn to empirical validation.

## 4 EXPERIMENTS

We evaluate RACE on widely used real-world benchmarks, the EuRoC MAV dataset Burri et al. (2016), the TUM RGB-D dataset Sturm et al. (2012), and the TartanAir Wang et al. (2020) monocular from the CVPR 2020 SLAM challenge (all licensed under CC-BY 4.0). Comparisons include classical calibration, recent deep learning approaches, and a combination of both Schönberger & Frahm (2016); Hagemann et al. (2023); DeTone et al. (2018); Sarlin et al. (2020); Fang et al. (2022); Arandjelovic et al. (2016).

To stress-test robustness (Section 4.1), we conduct rigorous ablation studies across intrinsic parameters, evaluating both convergence speed and steady-state parameter error. We also measure runtime overhead to confirm real-time feasibility on embedded hardware. Integrating RACE into a full visual-SLAM pipeline and evaluating its effect on volumetric 3D reconstruction are left for future work.

**Implementation Details:** All experiments were run on a single CPU core, whereas the deep-learning baselines required multiple GPUs for training and inference. RACE requires no pre-training or labeled data and converges directly from a single unlabeled trajectory, highlighting its efficiency and ease of deployment. Across all datasets (Tables 1, 2, 3), the average per-frame overhead

Table 1: RACE calibration performance on the EuRoC MAV sequences. The ground-truth camera intrinsics for EuRoC are $(f_x, f_y, c_x, c_y) = (458.7,\ 457.3,\ 367.2,\ 248.4)$.

| Metric | MH_01 | MH_02 | MH_03 | MH_05 | V1–01 | V1–02 | V1–03 | V2–01 | V2–02 | V2–03 | Avg | Med |
|---|---|---|---|---|---|---|---|---|---|---|---|---|
| Frame RMS-RE 5% | 70 | 64 | 118 | 125 | 190 | 168 | 261 | 268 | 190 | 305 | 170.73 | **168** |
| Time RMS-RE 5% (s) | 3.50 | 3.20 | 5.90 | 6.25 | 9.50 | 8.40 | 13.05 | 13.40 | 9.50 | 15.25 | 8.54 | **8.40** |
| Frame RMS-RE 1% | 149 | 124 | 273 | 243 | 241 | 253 | 460 | 411 | 421 | 620 | 311.45 | **253.0** |
| Time RMS-RE 1% (s) | 7.45 | 6.20 | 13.65 | 12.15 | 12.05 | 12.65 | 23.00 | 20.55 | 21.05 | 31.30 | 15.60 | **12.65** |
| Min RE (px) | **0.01** | 0.02 | 0.02 | 0.03 | 0.03 | 0.11 | 0.02 | 0.01 | 0.02 | 0.07 | 0.04 | 0.02 |
| Avg RE (px) | 0.43 | 0.43 | 0.41 | 0.44 | 0.42 | 0.40 | 0.41 | 0.42 | 0.42 | 0.43 | 0.42 | **0.42** |
| Avg Compute (ms) | 9.97 | 10.15 | 9.00 | 7.68 | 8.32 | 5.99 | 4.48 | 8.40 | 7.78 | 5.41 | 7.68 | **7.78** |

Table 2: RACE calibration performance on the TUM RGB-D sequences. The ground-truth camera intrinsics for TUM RGB-D are $(f_x, f_y, c_x, c_y) = (517.3,\ 516.5,\ 318.6,\ 255.3)$.

| Metric | 360 | desk | desk2 | floor | room | xyz | rpy | plant | teddy | Avg | Med |
|---|---|---|---|---|---|---|---|---|---|---|---|
| Frame RMS-RE 5% | 241 | 131 | 204 | 80 | 188 | 108 | 312 | 136 | 126 | 169.56 | 136.0 |
| Time RMS-RE 5% (s) | 8.04 | 4.37 | 6.80 | 2.66 | 6.27 | 3.60 | 10.40 | 4.53 | 4.26 | 5.66 | 4.53 |
| Frame RMS-RE 1% | 622 | 301 | 483 | 192 | 441 | 240 | 610 | 296 | 271 | 384 | 301 |
| Time RMS-RE 1% (s) | 20.74 | 10.04 | 16.10 | 6.40 | 14.70 | 8.00 | 20.33 | 9.87 | 9.02 | 12.80 | 10.04 |
| Min RE (px) | 0.60 | 0.11 | 0.30 | 0.08 | 0.12 | **0.10** | 0.46 | 0.12 | 0.10 | 0.22 | 0.12 |
| Avg RE (px) | 0.43 | 0.42 | 0.42 | 0.40 | 0.47 | 0.44 | 0.43 | 0.45 | 0.46 | 0.44 | **0.43** |
| Avg Compute (ms) | 8.63 | 11.27 | 10.02 | 12.14 | 10.35 | 12.11 | 9.31 | 10.86 | 10.23 | 10.55 | 10.35 |

is only **8.53** ms, well within real-time constraints and significantly lower than GPU-based methods. Additional implementation details are provided in Appendix D.

**Metrics:** Unless otherwise noted, updates are performed at 30 Hz and intrinsic parameters are initialized with a 25% offset from ground truth. Following Hagemann et al. (2023), we use root mean square reprojection error (RMS-RE) as the primary accuracy metric. To assess convergence, we report the first frame index and wall-clock time at which RMS-RE falls below 5% and 1% of its initial RMS-reprojection error value. Runtime efficiency is measured as average per-frame compute. For stability, we report two complementary measures: (i) *Min RE*, the minimum RMS-RE achieved within a sequence, and (ii) *Avg RE*, the average RMS-RE over all frames after Min RE is first reached, which reflects post-convergence stability. To ensure comparability with prior work, we summarize results using the same convention as Hagemann et al. (2023): the *median* (Med) value across sequences is reported for Avg RE. At the same time, Min RE is taken as the *minimum* across all sequences in the dataset. This choice highlights both the typical stability achieved after convergence (via median Avg RE) and the best-case accuracy attainable (via Min RE), enabling a fair comparison to published baselines (Table 4). The specific values used for comparison are bolded in the results table.

**EuRoC:** We first evaluate our approach on the EuRoC MAV benchmark (Burri et al., 2016), a standard dataset for SLAM and calibration that combines high-frequency stereo imagery from a micro-aerial vehicle with aggressive 6-DoF motion and challenging illumination conditions. RACE establishes a new state-of-the-art for intrinsic calibration on EuRoC. Across all sequences (Table 1), it converges to within **5%** of ground truth in a median of **8.4 s** (168 frames), and reaches the stringent **1%** threshold in just **12.7 s** (253 frames). The estimator never diverges and remains stable under long trajectories (median Avg RE of **0.42 px** across sequences (Table 4)), and runs in real time with an average overhead of only **7.8 ms** per frame on a single CPU core. Most importantly, RACE achieves the lowest reprojection error ever reported on EuRoC. As shown in Table 4, it matches the best monocular baseline (DroidCalib) in median Avg RE (**0.42 px**) while dramatically reducing the minimum error by **93%** (**0.01 px** vs. 0.16 px).

**TUM-RGBD:** On the TUM-RGBD benchmark (Sturm et al., 2012), RACE achieves state-of-the-art calibration results, setting a new reference point under challenging handheld indoor conditions. While prior methods suffer from rolling-shutter artifacts and frequent failures, our estimator successfully calibrates *all nine* sequences. RACE achieves a median Avg RE of **0.43 px**, representing an **86%** reduction relative to DroidCalib (3.09 px) (Table 4). The median Min RE is cut by **93.3%** (**0.10 px** vs. 1.50 px). Despite these large accuracy gains, runtime remains real-time at **10.4 ms** per frame on a single CPU core (Table 2).

Table 3: RACE calibration performance on the TartanAir sequences. The ground-truth camera intrinsics for TartanAir are $(f_x, f_y, c_x, c_y) = (320, 320, 320, 240)$.

| Metric | MH_000 | MH_001 | MH_002 | MH_003 | MH_004 | MH_005 | MH_006 | MH_007 | Avg | Med |
|---|---|---|---|---|---|---|---|---|---|---|
| Frame RMS-RE 5% | 27 | 76 | 80 | 321 | 74 | 93 | 128 | 185 | 123 | 86.50 |
| Time RMS-RE 5% (s) | 0.89 | 2.51 | 2.64 | 10.59 | 2.44 | 3.07 | 4.22 | 6.11 | 4.06 | 2.86 |
| Frame RMS-RE 1% | 162 | 263 | 385 | 372 | 533 | 602 | 405 | 690 | 426.50 | 395 |
| Time RMS-RE 1% (s) | 5.35 | 8.68 | 12.71 | 12.28 | 17.59 | 19.87 | 13.37 | 22.77 | 14.08 | 13.04 |
| Min RE (px) | **0.37** | 1.96 | 1.40 | 5.26 | 1.30 | 1.38 | 1.17 | 0.64 | 1.69 | 1.34 |
| Avg RE (px) | 16.81 | 7.33 | 17.31 | 25.72 | 18.48 | 20.24 | 24.01 | 19.70 | 18.7 | **19.09** |
| Avg Compute (ms) | 7.10 | 6.94 | 8.63 | 8.24 | 6.80 | 9.23 | 6.56 | 5.27 | 7.35 | 7.02 |

Table 4: Comparison of calibration methods on the TartanAir, EuRoC, TUM-RGBD, and raw EuRoC benchmarks Hagemann et al. (2023). Across all sequences; we report Min RE and Avg RE, respectively, as taken from Table 1, 2, 3, 6. For raw EuRoC, results use the OpenCV radial-distortion model,(*) denotes the opencv camera model with two radial distortion parameters and (**) denotes the unified camera model. Boldface highlights the lowest median error per dataset.

| Dataset | Method | *Med* (median) (px) | *Min RE* (PE) |
|---|---|---|---|
| TartanAir | COLMAP + NetVLAD | 1.45 | 0.11 |
| | COLMAP + NetVLAD + Superpoint + SuperGlue | 0.45 | 0.19 |
| | SelfSup–Calib** | 18.3 | 5.00 |
| | DroidCalib | **0.23** | 0.08 |
| | **RACE** (ours) | 19.09 | 0.37 |
| EuRoC | COLMAP + NetVLAD | 1.77 | 0.38 |
| | COLMAP + NetVLAD + Superpoint + SuperGlue | 0.71 | 0.42 |
| | SelfSup–Calib** | 27.6 | 14.0 |
| | DroidCalib | 0.42 | 0.16 |
| | **RACE** (ours) | **0.42** | **0.01** |
| TUM | COLMAP + NetVLAD | 6.54 | 2.53 |
| | COLMAP + NetVLAD + Superpoint + SuperGlue | 4.10 | 1.66 |
| | SelfSup–Calib** | 29.7 | 17.6 |
| | DroidCalib | 3.09 | 1.50 |
| | **RACE** (ours) | **0.43** | **0.10** |
| EuRoC Raw | COLMAP + NetVLAD* | 3.66 | 2.03 |
| | COLMAP + NetVLAD + Superpoint + SuperGlue* | 3.48 | 0.66 |
| | SelfSup–Calib** | 10.8 | 1.63 |
| | DroidCalib** | 0.40 | 0.31 |
| | **RACE** (ours) | **0.42** | **0.29** |

**TartanAir.** Finally, we evaluate on the TartanAir benchmark (Wang et al., 2020), focusing on the monocular "Hard" sequences from the ECCV 2020 SLAM competition. Unlike EuRoC and TUM-RGBD, these synthetic sequences deliberately stress calibration with low illumination, fog, and repeated textures. On this dataset, RACE converges stably across all runs and maintains real-time efficiency (7.0 ms per frame on average). Still, the accuracy is reduced: the median Avg RE is **19.09 px**, and the median Min RE is **0.37 px** (Table 3). In comparison, DroidCalib attains 0.23 px Avg RE ((Table 4). This performance gap is consistent with our theoretical requirement of persistent excitation: long stretches of low parallax and weak feature diversity degrade the regressor $\Phi_t$. We provide a detailed per-sequence analysis in the Supplementary Sec. 1.3.1, where we show that error spikes coincide with frames lacking informative features (e.g., low-light or foggy segments). These results highlight that while RACE is robust in real-world data, handling persistent excitation failures in synthetic edge cases remains an important avenue for future work. To address this, we can adopt gated adaptation law: $\dot{\hat{\theta}}_t = \begin{cases} -\Gamma \, \Phi_t^\top \tilde{x}_t, & \text{if PE} > \delta, \\ 0, & \text{otherwise,} \end{cases}$, for $\delta > 0$, that pauses updates, ensures $\dot{V}(t)$ is bounded until sufficient PE conditions recovers B.2.

## 4.1 ABLATION STUDY

To empirically test our theoretical guarantees, we perform ablations on the EuRoC MAV benchmark. We focus on verifying in practice both global asymptotic convergence of intrinsic parameters (Theorem 1) and bounded error behavior under measurement noise (Theorem 2). Two core exper-

iments are reported in the main text: (i) sensitivity to the initial intrinsic offset and (ii) robustness to pixel-level measurement noise, both under the pinhole model without distortion. Each ablation varies a single factor while keeping all others fixed. Results with lens distortion parameters, along with extended analyses of adaptation gain selection, initial offset, measurement noise, PE-condition violations, and drift scenarios (thermal, plateau, and combined), are provided in Appendix C.

**(i) Initial-offset sensitivity:** To validate the global asymptotic stability and convergence guaranteed by Theorem 1, we stress-test RACE on the EuRoC sequence (MH_01_easy) by re-initializing the intrinsics with extreme offsets of 100% relative to ground truth (Fig. 1). Extended results across a broader range of offsets {25%, 50%, 100%, 200%} (see Fig. 6) and sequences are summarized in Table 7 and discussed in the appendix C.3. These results highlight that even under such severe perturbations, RACE converges to the true parameters with only a modest increase in convergence time. While this establishes RACE as a practical online calibration algorithm under the known-pose assumption, extending the approach to handle unknown pose remains an important direction for future work. In addition, continuous long-duration parameter drifts (thermal, plateaus (abrupt parameter jumps), and combined) are reported in the appendix C.7.

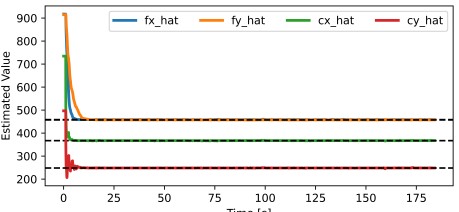

Figure 1: The MH_01 sequence was initialized with 100% offsets from the ground-truth intrinsics.

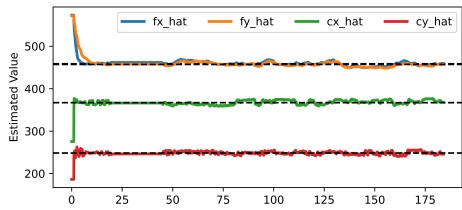

Figure 2: Parameter convergence on the MH_01 sequence under zero-mean Gaussian noise levels of 5 px.

**(ii) Measurement pixel noise.** We next test robustness to noisy observations by adding zero-mean Gaussian noise of 5 px to each reprojection measurement on MH_01. As shown in Fig. 2, the noisy runs closely track the noise-free baseline, differing only by a minor steady-state bias consistent with the bounded error guarantee of Theorem 2. Extended quantitative results across different noise ranges (Fig. 7) and sequences, along with analyses, are reported in the appendix C.4.

Finally, to verify that calibration improvements translate to downstream performance, we integrate RACE into a full visual odometry (ORB-SLAM3) and evaluate trajectory accuracy. RACE consistently improves post-convergence ATE compared to fixed or offline intrinsics (see Appendix F).

## 5 CONCLUSION

We presented RACE, a *truly online*, provably stable estimator for camera intrinsics that operates in real time on a single CPU core, without pre-collected calibration data, batch optimization, or GPU resources. The method is based on a lightweight control-theoretic update law that continually adapts intrinsic parameters. Our analysis provides unified guarantees: global stability, asymptotic convergence in the noise-free setting, and global uniform ultimate boundedness under bounded noise and persistent excitation. The framework extends directly to the distortion model. Empirically, RACE achieves sub-pixel RMS reprojection error across diverse benchmarks, setting a new state of the art in convergence speed while matching or surpassing the best classical and learning-based baselines in accuracy, all with negligible runtime overhead. In summary, RACE re-frames calibration as a control-theoretic online learning algorithm, enabling robust and continual adaptation for visual autonomy.

**Limitations and Future Work:** RACE assumes access to known 2D-3D correspondence and camera poses, and relies on persistent excitation to guarantee convergence and stability. While these assumptions are standard in calibration theory, they restrict applicability in settings with unknown poses. Extending RACE to operate robustly under unknown poses and weaker PE conditions

is an important direction for future work. Finally, generalizing the framework to multi-camera rigs, RGB-D sensors, or event cameras would broaden its applicability to a broader range of embodied AI systems.

## STATEMENTS

**Ethics Statement.** This work studies online estimation of camera intrinsics for embodied systems. Our experiments use only publicly available datasets (EuRoC MAV, TUM RGB-D, and TartanAir) under their respective terms of use; to our knowledge these datasets do not contain personally identifying information, and TUM RGB-D and TartanAir are distributed under CC BY 4.0 licenses (we follow their attribution requirements). We did not collect new human-subject data, nor did we recruit or interact with human participants. Potential dual-use risks include enabling more reliable perception for surveillance applications; to mitigate this, we evaluate only on established research datasets and release research-oriented code without any person-identification components. Our method assumes access to feature tracks and known poses (or a VO/SLAM subsystem) and should be validated by practitioners for safety before deployment on safety-critical platforms. We comply with dataset licenses/terms and the ICLR Code of Ethics.

**Reproducibility Statement.** We aim to make our results fully reproducible. The algorithmic update law and stability analysis are specified in the main text and Appendix A, with a step-by-step pseudo-algorithm provided in App. D.1 (Algorithm 1). Complete implementation details (feature extraction, PE gating, hyperparameters, and logging) are documented in App. D, and all ablation/VO experiments and metrics (including ATE via evo) are described in Sec. 4 and Apps. C & E, with per-sequence tables/plots and exact settings. An anonymous repository with code, configs, and scripts is linked in the abstract. It includes dependencies, dataset download helper scripts, run commands, and codes for all experiments. We also provide instructions to reproduce compute/runtime measurements and to regenerate all figures/tables from raw logs.

**LLM Usage.** We used a large language model (GPT 5 Thinking) only for minor editing (grammar and wording) after the technical content was finalized. The model did not propose ideas, proofs, experiments, code generation, or any literature survey. All research contributions and writing decisions are our own. We assume full responsibility for the content.

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

## A APPENDIX: STABILITY AND CONVERGENCE ANALYSIS (IDEAL NOISE FREE CASE

### A.1 LYAPUNOV STABILITY PROOF

We define a Lyapunov candidate function:

$$V(t) = \frac{1}{2}\tilde{\theta}_t^\top \Gamma^{-1} \tilde{\theta}_t, \tag{5}$$

where $\Gamma = \Gamma^\top \succ 0$ is the adaptation gain matrix. Differentiating $V(t)$ with respect to time yields:

$$\dot{V}(t) = \tilde{\theta}_t^\top \Gamma^{-1} \dot{\tilde{\theta}}_t. \tag{6}$$

Since the true intrinsics $\theta^\star$ are constant, we have $\dot{\tilde{\theta}}_t = \dot{\hat{\theta}}_t$. Also, $\tilde{p}_t = \Phi_t \tilde{\theta}_t$. Now substitute the adaptive update law:

$$\dot{\hat{\theta}}_t = -\Gamma \Phi_t^\top \tilde{p}_t, \tag{7}$$

which yields:

$$\dot{V}(t) = -\tilde{\theta}_t^\top \Phi_t^\top \tilde{p}_t. \tag{8}$$

Using $\tilde{p}_t = \Phi_t \tilde{\theta}_t$, we note that

$$\tilde{\theta}_t^\top \Phi_t^\top \tilde{p}_t = \tilde{p}_t^\top \tilde{p}_t.$$

Hence,

$$\dot{V}(t) = -\tilde{p}_t^\top \tilde{p}_t. \tag{9}$$

Therefore, the time derivative simplifies to:

$$\dot{V}(t) = -\tilde{p}_t^\top \tilde{p}_t \leq 0. \tag{10}$$

This guarantees that $V(t)$ is non-increasing over time and proves stability in the Lyapunov sense.

## A.2 Errors are bounded

Since the Lyapunov function satisfies $V(t) \geq 0$ and is non-increasing, i.e., $\dot{V}(t) \leq 0$. Consequently, both the parameter error $\tilde{\theta}_t$ and the reprojection error $\tilde{p}_t$ remain bounded for all $t \geq 0$. In particular,

$$\tilde{\theta}_t, \tilde{p}_t \in \mathcal{L}_\infty, \quad \text{that is,} \quad \|\tilde{\theta}_t\| < \infty, \quad \|\tilde{p}_t\| < \infty \quad \text{for all } t \geq 0.$$

## A.3 Reprojection error is $\mathcal{L}_2$

Integrating equation 10 from $t = 0$ to $t = \infty$ yields

$$V(\infty) - V(0) = -\int_0^\infty \tilde{p}_t^\top \tilde{p}_t \, dt. \tag{11}$$

Since $V(t)$ is non-increasing and lower bounded by zero, we have $V(\infty) \geq 0$ and $V(0) < \infty$, which implies

$$\int_0^\infty \|\tilde{p}_t\|^2 \, dt \leq V(0) < \infty. \tag{12}$$

Therefore, the reprojection error satisfies $\tilde{p}_t \in \mathcal{L}_2$.

## A.4 Uniform continuity

We now show that both $\tilde{p}_t$ and $\tilde{\theta}_t$ are uniformly continuous. We will use the standard result: *If a function has a bounded derivative, then it is uniformly continuous.*

We begin with the reprojection error dynamics:

$$\tilde{p}_t = \Phi_t \tilde{\theta}_t.$$

Differentiating the error dynamics yields the simplified expression:

$$\dot{\tilde{p}}_t = -\Phi_t \Gamma \Phi_t^\top \tilde{p}_t. \tag{13}$$

From the Lyapunov analysis, we have that $\Phi_t$, $\Gamma$, and $\tilde{p}_t$ are all bounded. Therefore, $\dot{\tilde{p}}_t$ is bounded, and it follows that $\tilde{p}_t$ is uniformly continuous.

**Uniform continuity of $\tilde{\theta}_t$.** From the adaptive update law $\dot{\tilde{\theta}}_t = -\Gamma \Phi_t^\top \tilde{p}_t$, we have

$$\dot{\tilde{\theta}}_t = -\Gamma \Phi_t^\top \tilde{p}_t.$$

Since $\Gamma$, $\Phi_t$, and $\tilde{p}_t$ are all bounded, it follows that $\dot{\tilde{\theta}}_t \in \mathcal{L}_\infty$. Hence, $\tilde{\theta}_t$ is uniformly continuous.

## A.5 Convergence of reprojection error

From the previous results, we have established that $\tilde{p}_t \in \mathcal{L}_2 \cap \mathcal{L}_\infty$ and that $\tilde{p}_t$ is uniformly continuous.

By applying *Barbalat's Lemma* B.1, which states that if a function $f(t)$ is uniformly continuous and satisfies $f \in \mathcal{L}_2 \cap \mathcal{L}_\infty$, then $f(t) \to 0$ as $t \to \infty$, we conclude:

$$\tilde{p}_t \to 0 \quad \text{as} \quad t \to \infty.$$

## A.6 Convergence of parameter under PE

**Convergence of the parameter error $\tilde{\theta}_t$.** Recall that $\tilde{p}_t = \Phi_t \tilde{\theta}_t$ and that $\tilde{p}_t \to 0$ as $t \to \infty$ by Barbalat's Lemma.

Assume that the regressor $\Phi_t \in \mathbb{R}^{2N(t) \times 4}$ satisfies the *persistent excitation (PE)* condition: there exist constants $\alpha > 0$ and $T > 0$ such that

$$\int_t^{t+T} \Phi_\tau^\top \Phi_\tau \, d\tau \succeq \alpha I_4, \quad \forall t \geq 0. \tag{14}$$

Using the relation $\tilde{p}_t = \Phi_t \tilde{\theta}_t$, we write:

$$\lim_{t \to \infty} \int_t^{t+T} \tilde{\theta}_\tau^\top \Phi_\tau^\top \Phi_\tau \tilde{\theta}_\tau \, d\tau = 0.$$

By the PE condition equation 14, it follows that

$$\alpha \int_t^{t+T} \|\tilde{\theta}_\tau\|^2 \, d\tau \leq \int_t^{t+T} \tilde{\theta}_\tau^\top \Phi_\tau^\top \Phi_\tau \tilde{\theta}_\tau \, d\tau \to 0,$$

so we conclude:

$$\int_t^{t+T} \|\tilde{\theta}_\tau\|^2 \, d\tau \to 0 \quad \text{as } t \to \infty. \tag{15}$$

Finally, since $\dot{\tilde{\theta}}_t = -\Gamma \Phi_t^\top \tilde{p}_t$ is bounded, we have that $\tilde{\theta}_t$ is uniformly continuous. Combining this with equation 15, we conclude that

$$\tilde{\theta}_t \to 0 \quad \text{as} \quad t \to \infty.$$

## B    BACKGROUND ON LYAPUNOV STABILITY AND ADAPTIVE CONTROL

### B.1    KEY THEOREMS

We here recall the key facts from continuous-time adaptive control that underlie our proofs:

- A function $V(x)$ is a **Lyapunov candidate** on a region $\mathcal{D}$ if $V(x) > 0$ in $\mathcal{D} \setminus \{0\}$ and $V(0) = 0$. If its time-derivative $\dot{V}(x) \leq 0$, then the equilibrium $x = 0$ is *stable* in the sense of Lyapunov (see Khalil & Grizzle (2002); Slotine et al. (1991)).
- **Barbalat's Lemma:** If $f(t)$ is uniformly continuous and $\int_0^\infty f^2(t) \, dt < \infty$, then $f(t) \to 0$. This lets us promote $\mathcal{L}_2 \cap \mathcal{L}_\infty$ to asymptotic convergenceSlotine et al. (1991).
- **Persistent Excitation:** A time-varying regressor $\Phi(t)$ is PE if $\int_t^{t+T} \Phi^\top \Phi \, d\tau \succeq \alpha I$ for some $\alpha, T > 0$. Under PE, an adaptive law of the form $\dot{\tilde{\theta}} = -\Gamma \Phi^\top (\Phi \tilde{\theta})$ will drive $\tilde{\theta} \to 0$ (cf.Slotine et al. (1991)).

### B.2    CLASSICAL GRADIENT ESTIMATOR AND PE CONDITION

For completeness, we recall the classical result for the gradient estimator Ioannou & Sun (1996); Narendra & Annaswamy (2012); Sastry & Bodson (2011), re-writing the the parameter error dynamics $\dot{\tilde{\theta}}_t = -\Gamma \Phi_t^\top \tilde{p}_t$,, with bounded regressor $\Phi_t$. Then:

1. $\|\tilde{\theta}(t_b)\| \leq \|\tilde{\theta}(t_a)\|$ for all $t_b \geq t_a$, the parameter error norm is monotonically non-increasing.
2. $\phi^\top(t)\tilde{\theta}(t) \to 0$ as $t \to \infty$.
3. The origin is globally exponentially stable iff $\phi \in$ PE.
4. Under PE, an optimal $\gamma$ exists that maximizes the convergence rate Ortega et al. (2020).

### B.3    LENS DISTORTION CAMERA MODEL

We use the classical Brown–Conrady radial–tangential model Conrady (1919). We have defined, $X_c = [X, Y, Z]$, now let $x_n = X/Z$, $y_n = Y/Z$, $r^2 = x_n^2 + y_n^2$, and $\mathbf{d} = [k_1, k_2, k_3, p_1, p_2]^\top$. With the first two radial terms (and optional $k_3$), the distorted normalized coordinates are

$$x_d = x_n\left(1 + k_1 r^2 + k_2 r^4\right) + 2p_1 x_n y_n + p_2\left(r^2 + 2x_n^2\right), \tag{16}$$

$$y_d = y_n\left(1 + k_1 r^2 + k_2 r^4\right) + p_1\left(r^2 + 2y_n^2\right) + 2p_2 x_n y_n, \tag{17}$$

and the distorted pixel projection is

$$u_d = f_x x_d + s\, y_d + c_x, \qquad v_d = f_y y_d + c_y, \tag{18}$$

with $s=0$ in our simulations and experiments. Higher-order/rational terms can be added for wide-FOV lenses; we retain the above for consistency with our evaluation.

Table 5: Evaluation of gain matrix selection and robustness on the EuRoC MAV sequence MH_01. For each run, intrinsic parameters are initialized with a 25% offset above ground truth values. We report: (i) the number of frames until the RMS-reprojection error (RE) falls below 5% and 1% of initial RMS-RE, (ii) the time taken to reach each of these thresholds, and (iii) Minimum reprojection error achieved at the end of the sequence. Results confirm that our selected diagonal gain matrix offers a robust balance between fast convergence and numerical stability.

| Diagonal Matrix | Frame RMS 5% | Time RMS 5% | Frame RMS 1% | Time RMS 1% | Min RE |
|---|---|---|---|---|---|
| $[10^{-5}, 10^{-5}, 20^{-5}, 20^{-5}]$ | - | - | - | - | 11.20 |
| $[10^{-4}, 10^{-4}, 20^{-4}, 20^{-4}]$ | - | - | - | - | 4.02 |
| $[10^{-3}, 10^{-3}, 20^{-3}, 20^{-3}]$ | 847 | 42 | 1394 | 69.70 | 0.03 |
| $[10^{-2}, 10^{-2}, 20^{-2}, 20^{-2}]$ | 70 | 3.50 | 149 | 7.55 | 0.01 |
| $[10^{-1}, 10^{-1}, 20^{-1}, 20^{-1}]$ | - | - | - | - | - |

# C    ABLATION STUDY

Beyond the main text, we include additional analyses in the appendix to further validate the robustness of RACE. These cover (i) adaptation gain selection, (ii) lens distortion parameters, (iii) extended offset and noise sweeps, (iv) Measurement Noise (v) PE-condition violations, and (vi) drift scenarios including thermal, plateau, and combined disturbances . Together, these experiments stress-test the our online learning algorithm under a wide spectrum of operating conditions and confirm that the theoretical guarantees extend to diverse real-world challenges.

## C.1    ADAPTION GAIN MATRIX SELECTION AND ROBUSTNESS

Prior to running experiments, we performed a sweep over several orders of magnitude to empirically select a gain that offers both stable and efficient convergence. We have added an ablation study (see Table 5), which demonstrates that the method remains robust across a broad range of gain values (from $10^{-5}$ to $10^{-2}$). It illustrates how the convergence rate of our method can be tuned to meet different application requirements.

As seen in the Table 5, smaller gain values lead to slower convergence. For example, with a gain matrix on the order of $10^{-5}$, the RMS-RE falls to approx. 18% of the initial RMS-RE by the end of the sequence. With $10^{-4}$, this error drops below 5.6%. While convergence is slower with smaller gains, longer trajectories guarantee that the estimate eventually approaches the true intrinsic parameters. In contrast, excessively high gain values (e.g., $10^{-1}$) can destabilize the estimation process by amplifying noise, resulting in divergence. Our selected gain matrix $[10^{-2}, 10^{-2}, 20^{-2}, 20^{-2}]$ offers a robust trade-off, achieving fast convergence while maintaining numerical stability across all evaluated datasets and sequences.

## C.2    DISTORTION CAMERA MODEL

We further evaluate *RACE* on EuRoC sequences synthetically re-rendered with a strong radial distortion (Table 6). The added nonlinearity slows convergence, median time to reach the 5% error band increases from **8.4 s** to **19.8 s** ($\times 2.4$) but the estimator remains stable and ultimately achieves the same post convergence accuracy as in the undistorted case (Avg RE $= 0.42$ px, cf. Table 1, Table 4).

The Min RE degrades from 0.01px to 0.29px, reflecting an irreducible bias from unmodeled higher-order distortion terms. Nonetheless, with strong distortion applied, RACE still improves over the best baseline by **6.5%** (0.29 px vs. 0.31 px). Interestingly, the average per-frame cost drops from **7.78 ms** to **4.63 ms**, since severe distortion reduces the number of reliable features.

Overall, these results confirm that the distortion-aware extension of RACE preserves real-time efficiency and delivers state-of-the-art accuracy even under significant optical aberrations.

Table 6: RACE calibration performance on Raw EuRoC MAV sequences (distortion included).

| Metric | MH_01 | MH_02 | MH_03 | MH_04 | MH_05 | V1–01 | V2–01 | V2–02 | Avg | Med |
|---|---|---|---|---|---|---|---|---|---|---|
| Frame RMS 5% | 275 | 256 | 367 | 465 | 425 | 319 | 493 | 460 | 382.5 | 396.0 |
| Time RMS 5% (s) | 13.75 | 12.80 | 18.35 | 23.25 | 21.25 | 15.95 | 24.65 | 23.00 | 19.12 | 19.80 |
| Frame RMS 1% | 1816 | 1747 | 2018 | 2063 | 1801 | 2569 | 1673 | 1768 | 1931.87 | 1808.5 |
| Time RMS 1% (s) | 90.80 | 87.35 | 100.90 | 103.15 | 90.05 | 128.45 | 83.65 | 88.40 | 96.59 | 90.42 |
| Min RE (px) | 0.34 | 0.36 | **0.29** | 0.29 | 0.39 | 0.34 | 0.39 | 0.36 | 0.35 | 0.35 |
| Avg RE (px) | 0.47 | 0.41 | 0.46 | 0.41 | 0.39 | 0.41 | 0.42 | 0.42 | 0.43 | **0.42** |
| Avg Compute (ms) | 4.52 | 4.47 | 7.32 | 4.34 | 7.22 | 7.55 | 4.48 | 4.75 | 5.59 | 4.63 |

Table 7: Initial-offset analysis across multiple sequences MH_01, V1_01, and V2_01. For each sequence, we initialize all intrinsics at offsets of {25 %, 50 %, 100 %, 200 %} more than the ground truth and measure: We report: (i) the number of frames until the RMS-reprojection error (RE) falls below 5% and 1% of initial RMS-RE, (ii) the time taken to reach each of these thresholds, (iii) Minimum reprojection error achieved at the end of the sequence, and (iv) Compute (ms): the average per-frame processing time on a single CPU core. These results demonstrate that RACE maintains sub-pixel accuracy and converges in tens to hundreds of frames even under severe miscalibration, confirming the global stability properties of Theorem 1.

| Dataset | Initial-offset | Frame RMS 5% | Time RMS 5% | Frame RMS 1% | Time RMS 1% | Avg Compute |
|---|---|---|---|---|---|---|
| MH_01 | 25% | 70 | 3.50 | 149 | 7.45 | 9.97 |
| | 50% | 104 | 5.20 | 172 | 8.60 | 10.00 |
| | 100% | 144 | 7.20 | 188 | 9.40 | 9.91 |
| | 200% | 168 | 8.40 | 225 | 11.25 | 10.02 |
| V1_01 | 25% | 190 | 9.50 | 241 | 12.05 | 8.32 |
| | 50% | 214 | 10.70 | 256 | 12.80 | 8.58 |
| | 100% | 235 | 11.75 | 278 | 13.90 | 8.51 |
| | 200% | 251 | 12.55 | 293 | 14.65 | 8.68 |
| V2_01 | 25% | 268 | 13.40 | 411 | 20.55 | 8.40 |
| | 50% | 320 | 16.00 | 472 | 23.60 | 7.53 |
| | 100% | 391 | 19.55 | 504 | 25.20 | 7.44 |
| | 200% | 461 | 23.05 | 537 | 26.85 | 8.62 |

## C.3 INITIAL-OFFSET

We evaluate RACE's convergence under large miscalibrations by initializing all intrinsics at {25%, 50%, 100%, 200%} more than the ground truth values. Figure 6 shows results on sequence MH_01, with extended per-sequence figures provided in Supplementary Sec. 1.1.1. On MH_01 and V1_01, RACE consistently drives the error below 1% of the ground truth intrinsics in under 15s even when starting with a 200% offset. On V2_01, which features less feature-rich motion, the 1% threshold is reached in approximately 30s (see Table 7). As expected, larger initial errors require more frames to correct, but the degradation is graceful: convergence time scales sublinearly with offset magnitude. This behavior confirms the global uniform stability of Theorem 1 and demonstrates that RACE can recover from severe miscalibration up to three-fold errors or more without manual reinitialization.

In addition, to evaluate robustness under independent parameter perturbations, we conducted 10 trials of RACE on EuRoC MH_01 sequence. In each trial, the initial intrinsic parameters were independently perturbed by sampling a random offset within {−30%, +30%} of each ground-truth parameter, producing different and uncorrelated initial intrinsic values.

Across all trials, RACE consistently converged to the correct intrinsics. Table 8 reports the mean ± standard deviation across trials for all key metrics, including: (i) the minimum reprojection error, (ii) the time required for the parameters error to fall below 5% and 1% of its initial value, (iii) the final intrinsic parameters values, and (iv) the final intrinsic parameter error $\tilde{\theta} = \|\hat{\theta}(t) - \theta^\star\|$.

These results demonstrate that RACE remains stable and reliable under independent perturbations of each intrinsic parameter, validating its robustness to realistic variations in initialization.

Table 8: Evaluation of RACE (10 trials) on EuRoC MH_01. Reported values are mean $\pm$ standard deviation. The ground truth intrinsic parameters are $\{f_x = 458.7, f_y = 457.3, c_x = 367.2, c_y = 248.4\}$

| Metric | Mean | Std |
|---|---|---|
| 5% intrinsic convergence time (s) | 2.555 | 0.708 |
| 1% intrinsic convergence time (s) | 5.785 | 1.521 |
| Minimum reprojection RMS (px) | 0.0051 | 0.0 |
| $F_x$ | 458.866 | 0.0 |
| $F_y$ | 457.374 | 0.0 |
| $c_x$ | 367.444 | 0.0 |
| $c_y$ | 248.370 | 0.0 |
| Final Intrinsic parameter error | 0.3 | 0.0 |

## C.4 ROBUSTNESS AGAINST NOISE

**Pixel Space**: To evaluate the effect of continuous pixel-level noise, we inject zero-mean Gaussian perturbations with standard deviations $\{3, 5\}$ px into the detected feature coordinates on real EuRoC sequences. As shown in Figure 7, (for extended sequence results, please refer to the Supplementary Sec. 1.1.4), the reprojection error exhibits only minor fluctuations and remains uniformly bounded throughout each trial, even under sustained noise injection.

**3D Landmarks**: We conducted and reported the ablation study where every 3D landmark in every per-frame of the sequence is perturbed with with i.i.d. Gaussian noise $\mathcal{N}(\mathbf{0}, \sigma^2 \mathbf{I}_3)$, where $\sigma \in \{1, 3, 5, 10\}$ cm along with 25% initial bias in parameters, see Table 9. The increased noise amplifies the reprojection error, and since our adaptive law scales the update proportionally to the reprojection error magnitude, this causes larger adaptation steps. This explains the observed faster initial convergence rate in high-noise scenarios. However, this does not imply that noise improves accuracy. As shown in 9, the final convergence precision degrades with higher noise levels, as expected.

## C.5 ROBUSTNESS AGAINST PE CONDITION VIOLATION

We conducted and reported extensive ablation studies EuRoC sequences to evaluate RACE's robustness under degraded PE conditions.

- In the first experiment, we randomly dropped 20% to 70% of features per-frame from the start of each sequence, simulating poor feature initialization. As shown in Table 10, RACE converged in all cases without divergence, and the time to reach both 5% and 1% intrinsic error thresholds grew gracefully with increasing dropout. This demonstrate high tolerance to feature sparsity.

- To stress-test the system further, we performed a more aggressive experiment where 20% to 90% of features were randomly dropped per frame starting 5 seconds into each sequence, emulating severe and inconsistent tracking conditions. Despite this dynamic degradation, the method continued to converge stably across EuRoC sequences, showcasing its robustness even under extreme feature loss scenarios, see Table 11.

- Finally, we evaluated the system under extremely degenerate motion by feeding a single fixed 2D–3D correspondence repeatedly over multiple iterations and still observed stable convergence of the intrinsic parameters, see Table 13.

## C.6 PE GATING

When evaluating RACE on the TartanAir benchmark, we observed that long stretches of low parallax and weak feature diversity degrade the regressor $\Phi_t$. As shown in Supplementary Sec. 1.3.1, the error spikes coincide with frames that lack informative features (e.g., low-light or foggy seg-

Table 9: Robustness of **RACE** to 3-D landmark noise on the EuRoC–MAV sequences `MH_01`, `V1_01`, and `V2_01`. All intrinsic parameters are *initialised* with a $+25\%$ bias relative to ground truth. At every frame we corrupt each camera-frame landmark $\mathbf{X}_c$ with i.i.d. Gaussian noise $\mathcal{N}(\mathbf{0}, \sigma^2 \mathbf{I}_3)$, where $\sigma \in \{1, 3, 5, 10\}$ cm. For each noise level, we report: (i) the number of frames until the RMS-reprojection error (RE) falls below 5% and 1% of initial RMS-RE, (ii) the time taken to reach each of these thresholds, (iii) Minimum reprojection error achieved at the end of the sequence. RACE converges in all cases and retains sub-pixel accuracy with no diverged trials, even at $\sigma = 5$ cm.

| Dataset | $\sigma$ (cm) | Frame 5 % | Time 5 % (s) | Frame 1 % | Time 1 % (s) | Min RE (px) |
|---------|------|-----------|--------------|-----------|--------------|-------------|
| MH_01 | 0 | 70 | 3.50 | 149 | 7.55 | 0.01 |
| | 1 | 71 | 3.55 | 141 | 7.05 | 0.59 |
| | 3 | 64 | 3.20 | 118 | 5.90 | 1.68 |
| | 5 | 55 | 2.75 | 1074 | 53.70 | 2.81 |
| | 10 | 6 | 0.30 | 1140 | 57.00 | 5.92 |
| V1_01 | 0 | 190 | 9.50 | 241 | 12.05 | 0.03 |
| | 1 | 185 | 9.25 | 238 | 11.90 | 0.46 |
| | 3 | 156 | 7.80 | 226 | 11.30 | 1.89 |
| | 5 | 102 | 5.10 | 201 | 10.05 | 3.83 |
| | 10 | 263 | 13.15 | 2311 | 115.55 | 11.75 |
| V2_01 | 0 | 268 | 13.40 | 411 | 20.55 | 0.01 |
| | 1 | 264 | 13.20 | 405 | 20.25 | 1.03 |
| | 3 | 242 | 12.10 | 368 | 18.40 | 2.94 |
| | 5 | 161 | 8.05 | 293 | 14.65 | 5.82 |
| | 10 | 419 | 20.95 | – | – | 11.46 |

ments).To address this, we proposed gated adaptation law: $\dot{\hat{\theta}}_t = \begin{cases} -\Gamma \, \Phi_t^\top \tilde{x}_t, & \text{if PE} > \delta, \\ 0, & \text{otherwise,} \end{cases}$, for $\delta > 0$, that pauses updates, ensures $\dot{V}(t)$ is bounded until sufficient PE conditions recovers.

In the initial TartanAir evaluation, we used a very small PE gating threshold of $PE \geq 0.1$, which allowed degraded updates (Table 3 and Supplementary Sec. 1.3.1). After re-running the TartanAir MH000 sequence with a stricter gate of $PE \geq 5$, we observed a clear reduction in both intrinsic parameter error and reprojection-error spikes, demonstrating the effectiveness of PE-gating (Figure 5). However, PE gating alone does not guarantee smooth convergence of $\hat{\theta} \to \theta^*$ because TartanAir contains intervals of low illumination and intermittent excitation, which produce severely ill-conditioned regressors $\Phi_t$, confirming our earlier assessment.

To further isolate the cause, and consistent with Appendix C.5, we selected a single TartanAir (MH000) frame satisfying: (i) $\geq$200 valid 2D–3D correspondences, and (ii) $PE \geq 0.9$. Running RACE for 25K static iterations on this fixed correspondence set (known pose) produced smooth convergence of both the intrinsics $\hat{\theta} \to \theta^*$ and the reprojection error $\tilde{p} \to 0$, as shown in Table 13. This confirms that when the regressor is well-conditioned even with low PE, and RACE ensure convergence. Thus, the failures on TartanAir arise primarily from low illumination frames making $\Phi_t$ ill-conditioned.

To demonstrate that RACE can reliably recover after a long period of weak PE, we conducted an additional ablation study on the EuRoC MH_01 sequence. We intentionally forced the PE value to remain below the gating threshold, freezing the parameter update from frames 50th to 500th. During this interval, the intrinsic estimates remained stable without drifting. Once informative motion resumed, the updates reactivated automatically and the estimates re-converged to the true parameters, as shown in Figure 5.

Table 10: To evaluate the robustness of the PE condition we did ablation on feature-track density. For each EuRoC sequence, we randomly discard a fixed fraction of features $[20\%, 40\%, 50\%, 60\%, 70\%]$ *from the first frame onward*. The adaptive gains are identical to the main experiments. Despite severe feature loss (up to 70 %), **RACE** still converges without divergence; convergence time grows gracefully. (i) the number of frames until the RMS-reprojection error (RE) falls below 5% and 1% of initial RMS-RE, (ii) the time taken to reach each of these thresholds, (iii) Minimum reprojection error achieved at the end of the sequence.

| Dataset | Drop | Frame 5 % | Time 5 % (s) | Frame 1 % | Time 1 % (s) | Min RE (px) |
|---------|------|-----------|--------------|-----------|--------------|-------------|
| MH_01 | 0% | 70 | 3.50 | 149 | 7.55 | 0.01 |
| | 20% | 145 | 7.25 | 336 | 16.80 | 0.04 |
| | 40% | 255 | 12.75 | 965 | 48.25 | 0.04 |
| | 50% | 311 | 15.55 | 1090 | 54.50 | 0.03 |
| | 60% | 335 | 16.75 | 1089 | 54.45 | 0.03 |
| | 70% | 576 | 28.80 | 1255 | 62.75 | 0.03 |
| V1_01 | 0% | 190 | 9.50 | 241 | 12.05 | 0.03 |
| | 20% | 225 | 11.25 | 320 | 16.00 | 0.05 |
| | 40% | 292 | 14.60 | 526 | 26.30 | 0.04 |
| | 50% | 335 | 16.75 | 667 | 33.35 | 0.04 |
| | 60% | 335 | 16.75 | 716 | 35.80 | 0.06 |
| | 70% | 396 | 19.80 | 965 | 48.25 | 0.03 |
| V2_01 | 0% | 268 | 13.40 | 411 | 20.55 | 0.01 |
| | 20% | 381 | 19.05 | 586 | 29.30 | 0.02 |
| | 40% | 531 | 26.55 | 830 | 41.50 | 0.02 |
| | 50% | 540 | 27.00 | 865 | 43.25 | 0.02 |
| | 60% | 591 | 29.55 | 915 | 45.75 | 0.02 |
| | 70% | 665 | 33.25 | 1290 | 64.50 | 0.06 |

## C.7 DRIFT ROBUSTNESS

In addition to the controlled ablations reported in the main text, we evaluate RACE under more realistic long-horizon disturbances that arise in practice but are difficult to summarize in tables. Specifically, we analyze three scenarios:

1. *Thermal Drift*, modeling gradual sinusoidal changes in intrinsics due to heating. $\theta_t = \theta_0 \cdot \left(1 + A_{\text{therm}} \sin(2\pi t/T_{\text{therm}})\right)$, where $\theta_0$ is the ground truth intrinsic parameters. We consider a range of amplitude $A_{\text{therm}} = \{10, 20\}\%$ and $T_{\text{therm}} = 10s$ as the drift period, see Fig. 8.

2. *Plateau Drift*, representing abrupt step changes from shocks or hardware adjustments. To emulate this, we apply step changes of $\{5\%, 10\%, 20\%\}$ to all intrinsics at $t = \{30, 50, 80\}$ s, on top of a 25% initial offset, see Fig. 9.

3. *Combined disturbance* case where we simultaneously inject multiple sources of disturbance: (i) 25% initial intrinsic offset, (ii) additive continuous zero-mean pixel noise with 0.5 px, (iii) thermal drift of 10% peak amplitude with a 10 s period, and (iv) a plateau drift of 5% applied to all intrinsics at $t = \{30, 60, 90\}$s, see Fig.10.

Across all cases, RACE remains stable and consistently re-converges, maintaining sub-pixel reprojection error even under compounded disturbances. These experiments further validate the robustness predicted by our theoretical analysis and highlight the suitability of RACE for long-duration, real-world deployments. Extensive discussion and figures are illustrated in Supplementary.

## D IMPLEMENTATION DETAILS

All experiments run on a single 13th Gen Intel® Core™ i9-13980HX (no GPU) using Python and the following libraries:

Table 11: To evaluate the robustness of PE, we did an ablation on delayed feature loss. Each run starts with the full feature track set; after 5 sec we randomly discard a fixed fraction of feature tracks ($[20\%, 40\%, 50\%, 60\%, 70\%, 90\%]$ of the current set) and keep that reduced density for the remainder of the trajectory. (i) the number of frames until the RMS-reprojection error (RE) falls below 5% and 1% of initial RMS-RE, (ii) the time taken to reach each of these thresholds, (iii) Minimum reprojection error achieved at the end of the sequence. RACE degrades gracefully and remains stable even under extreme feature loss.

| Dataset | Drop % | Frame 5 % | Time 5 % (s) | Frame 1 % | Time 1 % (s) | Min RE (px) |
|---|---|---|---|---|---|---|
| MH_01 | 0% | 70 | 3.50 | 149 | 7.55 | 0.01 |
| | 20% | 72 | 3.60 | 200 | 10.00 | 0.03 |
| | 40% | 72 | 3.60 | 360 | 18.00 | 0.03 |
| | 50% | 72 | 3.60 | 430 | 21.50 | 0.03 |
| | 60% | 72 | 3.60 | 475 | 23.75 | 0.04 |
| | 70% | 72 | 3.60 | 808 | 40.40 | 0.03 |
| | 90% | 72 | 3.60 | 1317 | 65.85 | 0.09 |
| V1_01 | 0% | 190 | 9.50 | 241 | 12.05 | 0.03 |
| | 20% | 217 | 10.85 | 315 | 15.75 | 0.05 |
| | 40% | 273 | 13.65 | 496 | 24.80 | 0.04 |
| | 50% | 291 | 14.55 | 621 | 31.05 | 0.03 |
| | 60% | 300 | 15.00 | 640 | 32.00 | 0.05 |
| | 70% | 338 | 16.90 | 908 | 45.40 | 0.03 |
| | 90% | 747 | 37.35 | 2229 | 111.45 | 0.36 |
| V2_01 | 0% | 268 | 13.40 | 411 | 20.55 | 0.01 |
| | 20% | 375 | 18.75 | 585 | 29.25 | 0.02 |
| | 40% | 520 | 26.00 | 818 | 40.90 | 0.02 |
| | 50% | 525 | 26.25 | 842 | 42.10 | 0.01 |
| | 60% | 566 | 28.30 | 901 | 45.05 | 0.01 |
| | 70% | 630 | 31.50 | 1270 | 63.50 | 0.05 |
| | 90% | 1327 | 66.35 | - | - | 1.34 |

Table 12: Convergence of intrinsic parameters under degenerate motion using a single repeated 2D–3D correspondence. Despite the absence of Persistent Excitation, the reprojection error steadily decreases to zero. We report: (i) the RMS reprojection error after certain iterations.

| Iteration | 0 | 1000 | 2000 | 3000 | 4000 | 5000 | 6000 | 7000 | 8000 | 9000 |
|---|---|---|---|---|---|---|---|---|---|---|
| RMS-RE (px) | 72.33 | 3.05 | 0.86 | 0.24 | 0.07 | 0.02 | 0.01 | 0.00 | 0.00 | 0.00 |

- *opencv-python*, *numpy*, *scipy*, *pandas*, *PyYAML*, *matplotlib*, *tqdm*

The adaptation pipeline for one EuRoC sequence proceeds as follows:

1. **Configuration:** set *ROOT_DIR*, *MIN_TRACKS=30*, *SEED_EVERY=5*.

2. **Data loading:** parse intrinsics/extrinsics from YAML and poses from CSV; compute camera–world transforms.

3. **Initialization:** start $\hat{\theta}(0)$ at 25% offset; use gain $\Gamma = \mathrm{diag}(1\times10^{-2}, 1\times10^{-2}, 2\times10^{-2}, 2\times10^{-2})$; seed FAST features.

4. **Main loop (per frame):**

   - Detect new FAST points every *SEED_EVERY* frames.
   - Track features.
   - When $\geq$ *MIN_TRACKS* inliers exist, triangulate 3D points from the two most recent poses, compute reprojection residuals and Jacobians, and apply the continuous-time adaptation law:
   $$\hat{\theta} \leftarrow \hat{\theta} - \Gamma\Phi^{\top}\tilde{p}\Delta t$$

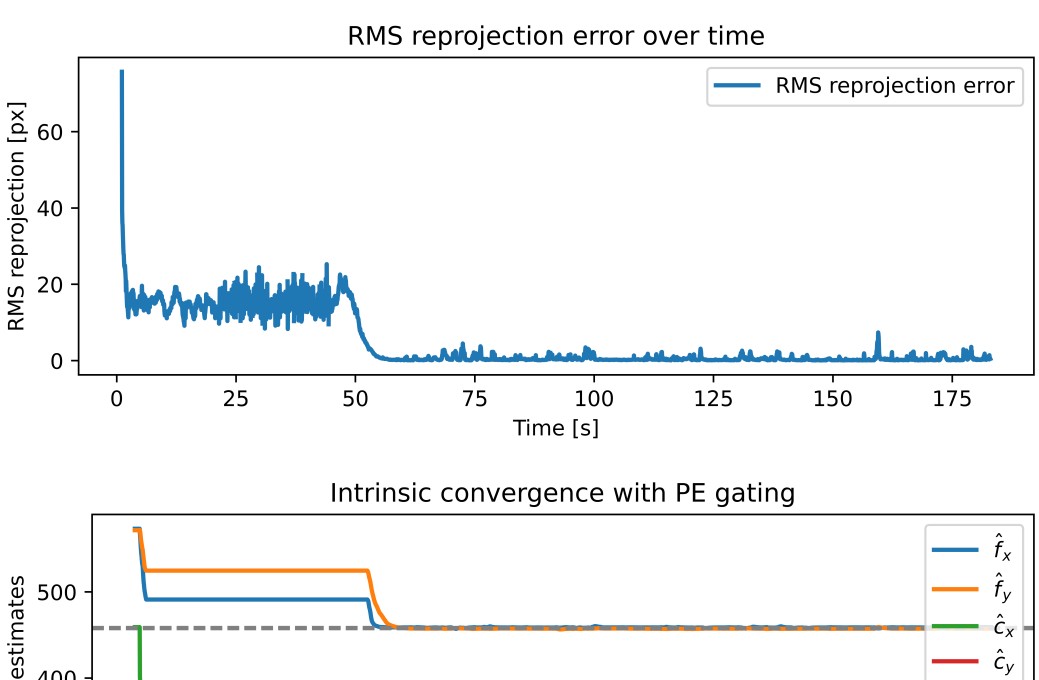

Figure 3: Effect of PE gating on the EuRoC MH_01 sequence. From frames 50th to 500th, the PE value is forced below the gating threshold, freezing the parameter update. **Top:** RMS reprojection error over time. **Bottom:** Convergence of intrinsic parameters, where each colored curve tracks one parameter and black dashed lines denote ground truth. During the frozen interval the intrinsics remain stable without drifting, and once informative motion resumes, RACE automatically reactivates and true parameters are recovered.

Table 13: The true camera intrinsics are $(f_x, f_y, c_x, c_y) = (320, \ 320, \ 320, \ 240)$, which RACE successfully recovers from a single repeated 2D–3D correspondence set (TartanAir MH000).

| Iteration | 0 | 5000 | 10000 | 15000 | 20000 | 25000 |
|---|---|---|---|---|---|---|
| RMS Error (px) | 28.87 | 1.26 | 0.23 | 0.04 | 0.01 | 0.00 |

- Log per-frame: timestamps, $\hat{\theta}$, track count, RMS reprojection error, parameter-error norm, smallest singular value of $H$, and processing time (ms).

5. **Output:** write per-frame root mean square projection error, theta error and summary CSV with convergence times, area-under-error curves, and runtime statistics.

## D.1 PSEUDO ALGORITHM

To make the estimator concrete, we summarize RACE as pseudo code in Algorithm 1. The procedure operates in a streaming fashion, updating intrinsics at every frame using reprojection errors and the regressor $\Phi_t$. The update is gated by the persistent excitation (PE) condition to avoid instability when feature diversity is insufficient. Distortion parameters $\mathbf{d}$ are optional; when included, they

---

**Algorithm 1** RACE: Real-Time Adaptive Camera-Intrinsic Estimation (Distortion-Optional)

---

1: **Inputs:** initial estimate $\hat{\theta}_0 = [f_x, f_y, c_x, c_y, \hat{\mathbf{d}}^\top]^\top$, adaptation gain $\Gamma \succ 0$, PE threshold $\delta > 0$, step size $\zeta$
2: **Assumptions:** known 2D–3D correspondences; camera pose $(R_t, \mathbf{t}_t)$ known
3: **while** frames arrive **do**
4:      Detect 2D features $\{\mathbf{p}_{i,t}^\star\}_{i=1}^{N_t}$ and collect 3D points $\{\mathbf{X}_{w,i}\}_{i=1}^{N_t}$
5:      Transform points to camera frame: $\mathbf{X}_{c,i} \leftarrow R_t \mathbf{X}_{w,i} + \mathbf{t}_t$
6:      Predict pixel locations: $\hat{\mathbf{p}}_{i,t} \leftarrow \pi(\mathbf{X}_{c,i}; \hat{\theta}_t, \hat{\mathbf{d}})$
7:      Compute reprojection error: $\tilde{\mathbf{p}}_{i,t} \leftarrow \hat{\mathbf{p}}_{i,t} - \mathbf{p}_{i,t}^\star$
8:      Form regressor $\Phi_t \in \mathbb{R}^{2N_t \times dim}$     $(dim = dim(\theta))$
9:      Stack inlier set: $(\tilde{\mathbf{p}}_t, \Phi_t)$
10:      **if** $\mathrm{PE}(\Phi_t) \geq \delta$ **then**
11:          Update: $\dot{\hat{\theta}}_t \leftarrow -\Gamma \Phi_t^\top \tilde{\mathbf{p}}_t$
12:          $\hat{\theta}_{t+1} \leftarrow \hat{\theta}_t + \zeta \dot{\hat{\theta}}_t$
13:      **else**
14:          Hold: $\hat{\theta}_{t+1} \leftarrow \hat{\theta}_t$
15:      **end if**
16:      **if** $\mathrm{RMS}(\tilde{\mathbf{p}}_t) < \varepsilon$ **then**
17:          **Converged**
18:      **end if**
19: **end while**
20: **Output:** estimated intrinsics $\theta^\star \leftarrow \hat{\theta}$ (and $\mathbf{d}^\star$ if enabled)

---

are seamlessly integrated into the parameter vector and initialized to zero. The projection function $\pi(\cdot; \theta)$ is the standard pinhole mapping from a 3D camera frame point $\mathbf{X}_c = (X, Y, Z)^\top$ to pixel coordinates $\mathbf{p}$ under intrinsics $\theta$.

## E   ADDITIONAL BASELINES

To contextualize the performance of RACE, we compare against three classical estimators commonly used for parameter identification: Least Squares (LS), Recursive Least Squares (RLS), and Gauss Newton (GN). Although these baselines can estimate intrinsic parameters in simplified settings, they lack the robustness, stability, and adaptability required for online calibration in realistic scenarios.

### E.1   LEAST SQUARES (LS)

LS is often considered for estimation because it can attenuate noise by solving a batch regression over many measurements Slotine et al. (1991). However, LS is fundamentally different from RACE for three key reasons:

1. **RACE is recursive and causal:** RACE updates parameters per-frame in real time without solving a batch problem. In contrast, LS requires storing all past data (or a growing window) and re-solving a global optimization at each step.

2. **RACE has formal stability and convergence guarantees:** The RACE update is derived from a Lyapunov function and satisfies theorems ensuring stability and boundedness (Theorems 1 and 2).

3. **RACE explicitly handles time-varying intrinsics:** RACE guarantees parameter tracking under time varying intrinsics. LS implicitly assumes static parameters and therefore fails when intrinsics drift or change over time.

These differences are clearly demonstrated in the *time-varying intrinsics* experiment (Table 14). LS diverges rapidly and shows no ability to track continuously changing parameters. In contrast, RACE continuously minimizes the reprojection error at every frame and reliably tracks the intrinsic

Table 14: Performance comparison between RACE and Least Squares (LS) under time varying intrinsic parameters. We report the final RMS reprojection error at the end of the EuRoC MH_01_easy sequence. RACE reliably tracks the drifting intrinsics, whereas LS fails to do so, resulting in significantly higher reprojection error.

| Method | Final RMS [px] |
|--------|----------------|
| RACE   | **6.6**        |
| LS     | 47.2           |

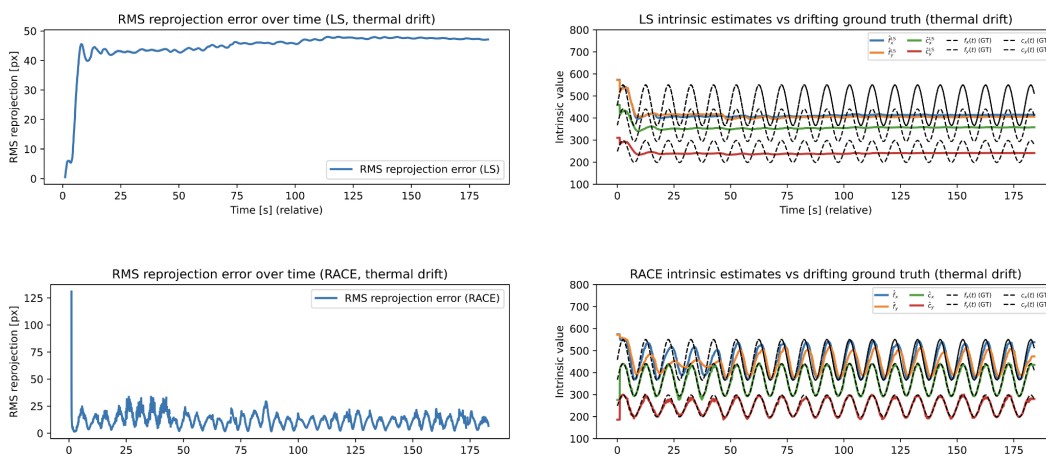

Figure 4: Comparison of LS and RACE on the EuRoC MH_01_easy sequence. The figure contains four panels: LS (top row) and RACE (bottom row), each showing (i) RMS reprojection error over time and (ii) intrinsic parameter tracking. LS fails to reliably track the intrinsic parameters and exhibits higher reprojection error, whereas RACE maintains low reprojection error and accurately follows the intrinsic drift throughout the sequence.

parameters as they evolve over time, maintaining precise alignment with the drifting ground truth, as illustrated in Figure 4.

### E.2 RECURSIVE LEAST SQUARES (RLS)

RLS provides a recursive alternative to LS, but its performance depends heavily on the forgetting factor $\lambda$. Our experiments show that: RLS is highly sensitive to the factor $\lambda$, small changes ($\lambda = 0.90$) lead to large variations in both reprojection and parameter error, and the method becomes unstable as noise increases, Table 15.

### E.3 GAUSS NEWTON (GN)

Online GN suffers from severe step size sensitivity: large steps overshoot or oscillate, while small steps converge slowly. Moderate step sizes (e.g., $\alpha = 2.0$) cause divergence with intrinsic parameter errors exceeding 200–300 pixels, and GN remains sensitive to noise even when tuned. GN offers no guarantees on stability or boundedness.

**Summary.** Table 15 reports intrinsic parameter error and RMS reprojection error across baselines. RACE remains stable, convergent, and robust to noise and time-varying intrinsics, whereas LS, RLS, and GN all fail under at least one of these conditions.

Table 15: Ablation over hyperparameters and pixel noise. For each method, we sweep key hyperparameters and evaluate at different pixel noise levels $\sigma$. We report the **final RMS reprojection and intrinsics parameter error** obtained at the end of the sequence.

| Method | Hyperparameter | $\sigma$ [px] | Final RMS [px] | Final Parameter Error [px] |
|---|---|---|---|---|
| **RLS** | | | | |
| RLS | $\lambda = 0.90$ | 0 | 0.44 | 0.57 |
| RLS | $\lambda = 0.95$ | 0 | 0.43 | 0.59 |
| RLS | $\lambda = 0.99$ | 0 | 0.42 | 0.77 |
| RLS | $\lambda = 0.90$ | 5 | 5.8 | 15 |
| RLS | $\lambda = 0.95$ | 5 | 5.7 | 4.4 |
| RLS | $\lambda = 0.99$ | 5 | 5.2 | 9.2 |
| RLS | $\lambda = 0.90$ | 10 | 10.8 | 24.96 |
| RLS | $\lambda = 0.95$ | 10 | 10.5 | 21.7 |
| RLS | $\lambda = 0.99$ | 10 | 10 | 13.2 |
| **GN (varying step size $\alpha$)** | | | | |
| GN | $\alpha = 0.1$ | 0 | 0.44 | 0.49 |
| GN | $\alpha = 0.3$ | 0 | 0.43 | 0.81 |
| GN | $\alpha = 1.0$ | 0 | 0.42 | 0.6 |
| GN | $\alpha = 2.0$ | 0 | 67.4 | 215.9 |
| GN | $\alpha = 0.1$ | 5 | 5.1 | 0.6 |
| GN | $\alpha = 0.3$ | 5 | 5.1 | 1.1 |
| GN | $\alpha = 1.0$ | 5 | 5.1 | 0.51 |
| GN | $\alpha = 2.0$ | 5 | 63 | 226.5 |
| GN | $\alpha = 0.1$ | 10 | 10 | 0.68 |
| GN | $\alpha = 0.3$ | 10 | 10.4 | 1.1 |
| GN | $\alpha = 1.0$ | 10 | 10 | 3.1 |
| GN | $\alpha = 2$ | 10 | 75.2 | 335.8 |
| **RACE ($\Gamma$)** | | | | |
| RACE | $10^{-2}$ | 0 | 0.43 | 0.32 |
| RACE | $10^{-2}$ | 5 | 4.6 | 3.4 |
| RACE | $10^{-2}$ | 10 | 8.9 | 8.2 |

## F  DOWNSTREAM VISUAL-ODOMETRY TESTS

For the EuRoC sequences Burri et al. (2016), we compare ORB-SLAM3 using fixed (ground-truth) intrinsics against ORB-SLAM3 using intrinsics updated online by RACE, initialized with a 25% error. We evaluate ATE Sturm et al. (2012) using `evo` Grupp (2017) and report two metrics; *Full* and *Segment* ATE (Table 16). *Full* corresponds to the ATE over the entire sequence, whereas *Segment* corresponds to the ATE obtained by re-running ORB-SLAM3 after the intrinsic parameters have converged.

As expected, *Full* ATE can be worse for RACE due to early miscalibration. However, the *Segment* results demonstrate the benefits after convergence. In sequences affected by temperature or focus changes, RACE yields lower *Segment* ATE than fixed/offline intrinsics, while adding negligible runtime overhead.

**Further Clarification for ORBSLAM3 setup:** RACE is first run on the trajectory using ground truth poses and initial intrinsics with a +25% bias to estimate per-frame time series of intrinsic parameters. The landmarks used for RACE are triangulated using ground truth poses and initial intrinsics, and not from the SLAM mapping. Each image frame (in the time series) is then warped using the corresponding intrinsic time series, effectively simulating as if the images were captured with incorrect calibration parameters. The warping method used is a OpenCV function - warpPerspective. ORB-SLAM3 is then run on this preprocessed warped sequence without modifying its internal calibration or tracking pipeline. The "Full" trajectory (in Table 12) has an higher error in RACE with preprocessed warped images since ORB-SLAM3 running on warped images results in higher error in the beginning. As the images become less warped due to intrinsic convergence, the

trajectory error starts reducing resulting in lower error in the "Segment" error. The "Segment" error is the trajectory error observed after the intrinsic parameters have converged to the true value.

Table 16: EuRoC VO accuracy (ATE RMSE) for 4 different scenarios. Trajectories are aligned/scaled with `evo` Grupp (2017). Segment window is the common tail $[t_{\text{conv}}, T]$ from time of convergence to end of the sequence determined by RACE; same tail used for both methods.

| Seq. | Fixed Intrinsics | | **RACE (online)** | |
|------|------|---------|------|---------|
|      | Full | Segment | Full | Segment |
| MH_01 | 0.016 | 0.032 | 0.216 | **0.029** |
| MH_02 | 0.027 | **0.031** | 0.086 | 0.053 |
| MH_03 | 0.028 | **0.033** | 0.409 | 0.042 |
| MH_04 | 0.138 | **0.042** | 0.346 | 0.047 |
| MH_05 | 0.072 | 0.049 | 0.823 | **0.034** |
| V1_01 | 0.033 | 0.081 | 0.434 | **0.079** |
| V1_02 | 0.015 | 0.063 | 0.987 | **0.043** |
| V1_03 | 0.033 | 0.066 | 0.976 | **0.061** |
| V2_01 | 0.023 | 0.060 | 0.618 | **0.028** |
| V2_02 | 0.029 | 0.032 | 0.332 | **0.021** |
| V2_03 | – | 0.933 | 0.768 | **0.542** |
| **Avg** | 0.041 | 0.129 | 0.545 | **0.089** |

# G  RACE-THEORETICALLY GROUNDED ONLINE CAMERA CALIBRATION METHOD AND FUTURE DIRECTION

Initially, our formulation assumes access to known poses and accurate 2D-3D correspondences. This assumption was made intentionally so that we could isolate and analyze the intrinsic parameter convergence and stability properties of RACE. Our immediate goal is to establish RACE as a theoretically grounded online camera calibration method, by relaxing know pose assumption.

Building on Appendix C.5(intrinsic estimation from a single frame), we show that the "known pose" assumption can be relaxed in several practical ways. One approach is to estimate the camera pose analytically using a calibration target (with known 2D-3D correspondences, as in Zhang (2000)), and then apply RACE to refine the intrinsics.

To evaluate this idea, we developed a simulation setup that mirrors this real world scenario. A virtual calibration board containing $25 \times 10$ planar points, and a fixed camera emulate a target based calibration setup.

We begin by analytically computing an initial estimate of both the extrinsic and intrinsic parameters from a single image of the calibration board. Keeping the estimated pose fixed, we then apply RACE to iteratively adapt the intrinsic parameters. Next, using these updated intrinsics, we refine the pose and run RACE again to update the intrinsics with the refined pose. In total, we perform a single cycle of alternating intrinsic and extrinsic refinement.

This procedure enables full intrinsic calibration from just one frame. As summarized in Table 17, the resulting accuracy is competitive with standard offline calibration methods. The only assumption required is that the calibration board is placed at a known depth from the camera, a quantity that can be easily obtained using modern RGB-D sensors in real world setting.

**Future Direction**: After relaxing the pose requirement, the remaining assumption for real-world deployment is access to reliable 2D-3D correspondences. We envision addressing this using a lightweight learning based model that predicts correspondences directly from unconstrained images. Overall, this show the pathway of fusing the control theory guarantee with learning based generalization for online estimation of pose and camera calibration, showing the practicality of our method.

| Method | Estimated Intrinsics $(f_x, f_y, c_x, c_y)$ | Final RMS (px) | Parameter Error |
|---|---|---|---|
| **Zhang (1 image)** | $(767.54,\ 763.81,\ 320.00,\ 240.00)$ | 1.4 | 47.55 |
| **RACE (True pose)** | $(800.000,\ 799.999999,\ 320.00,\ 240.00)$ | 0.00 | 0.001 |
| **RACE (Est. pose)** | $(801.3,\ 796.8,\ 319.94,\ 240.4)$ | 0.14 | 3.46 |
| **Zhang (15 images)** | $(800.322,\ 800.337,\ 320.239,\ 240.373)$ | 0.495 | 0.661 |

Table 17: **Intrinsic calibration results across four methods.** The true camera intrinsics are $\theta_{\text{true}} = (f_x, f_y, c_x, c_y) = (800,\ 800,\ 320,\ 240)$. The table reports the estimated intrinsics for each method, the final RMS reprojection error, and the intrinsic parameter norm error $\|\theta_{\text{est}} - \theta_{\text{true}}\|_2$. Single-image Zhang calibration shows large deviations in focal lengths. RACE with the true pose converges exactly to the correct intrinsics. RACE using only the pose estimated from Zhang's single-image solution significantly improves the intrinsics, reducing the norm error from 47.55 to 3.46. A standard 15-image Zhang calibration with nonlinear optimization provides sub-pixel intrinsic accuracy and low reprojection error.

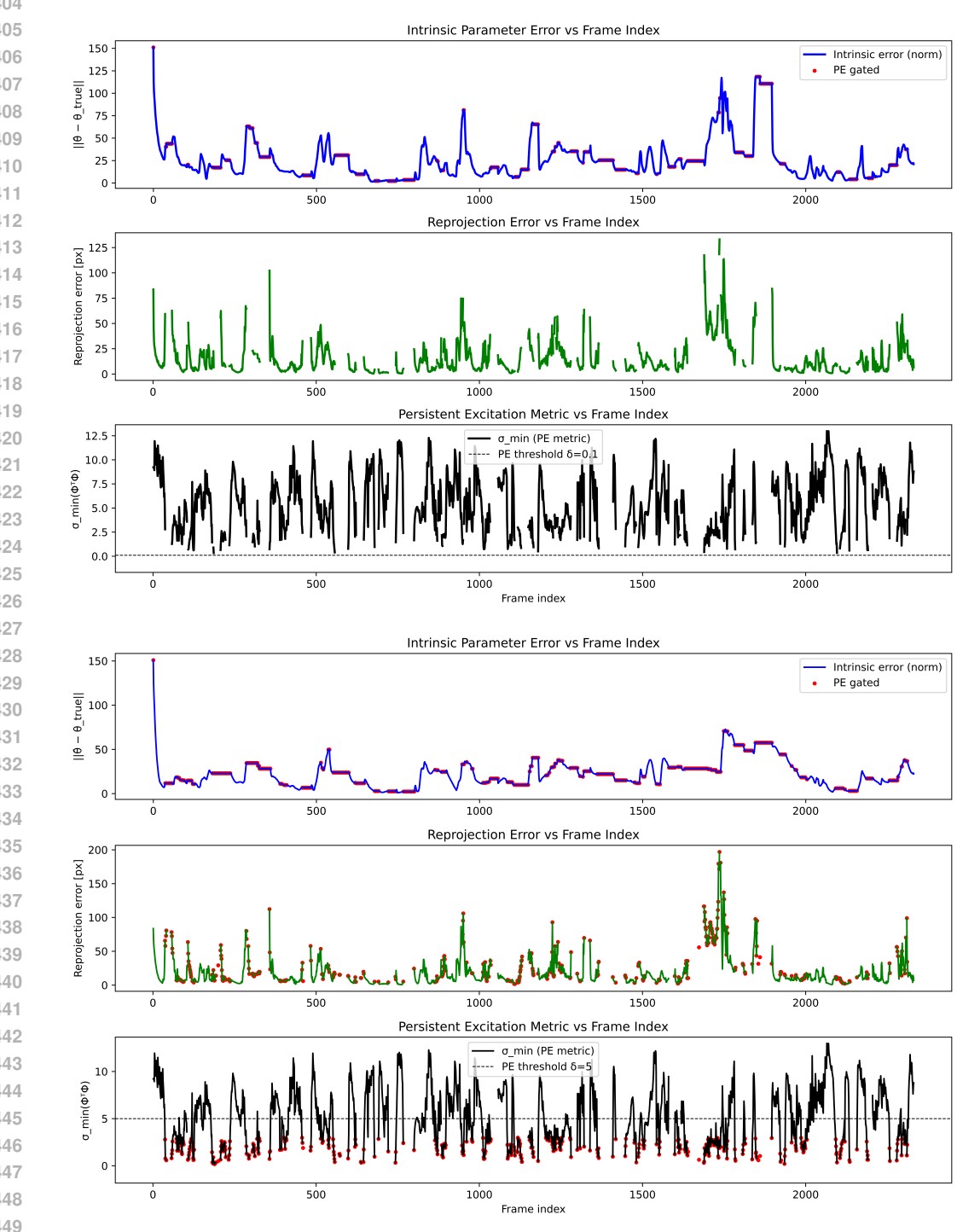

Figure 5: Effect of PE-gating on the TartanAir (MH000) sequence. **Top:** Our initial evaluation with a loose PE threshold ($PE \geq 0.1$), which allows degraded updates. **Bottom:** then apply a stricter gate ($PE \geq 5$). Red points in both plots denote frames where the PE condition is violated and the update is frozen. For each case, we reported, RMS reprojection error, intrinsics parameter error and PE values over frames. The black dashed lines indicate PE=gating Threshold. When PE falls below threshold, the update is frozen and the intrinsics remain stable; once informative motion resumes, RACE automatically reactivates.

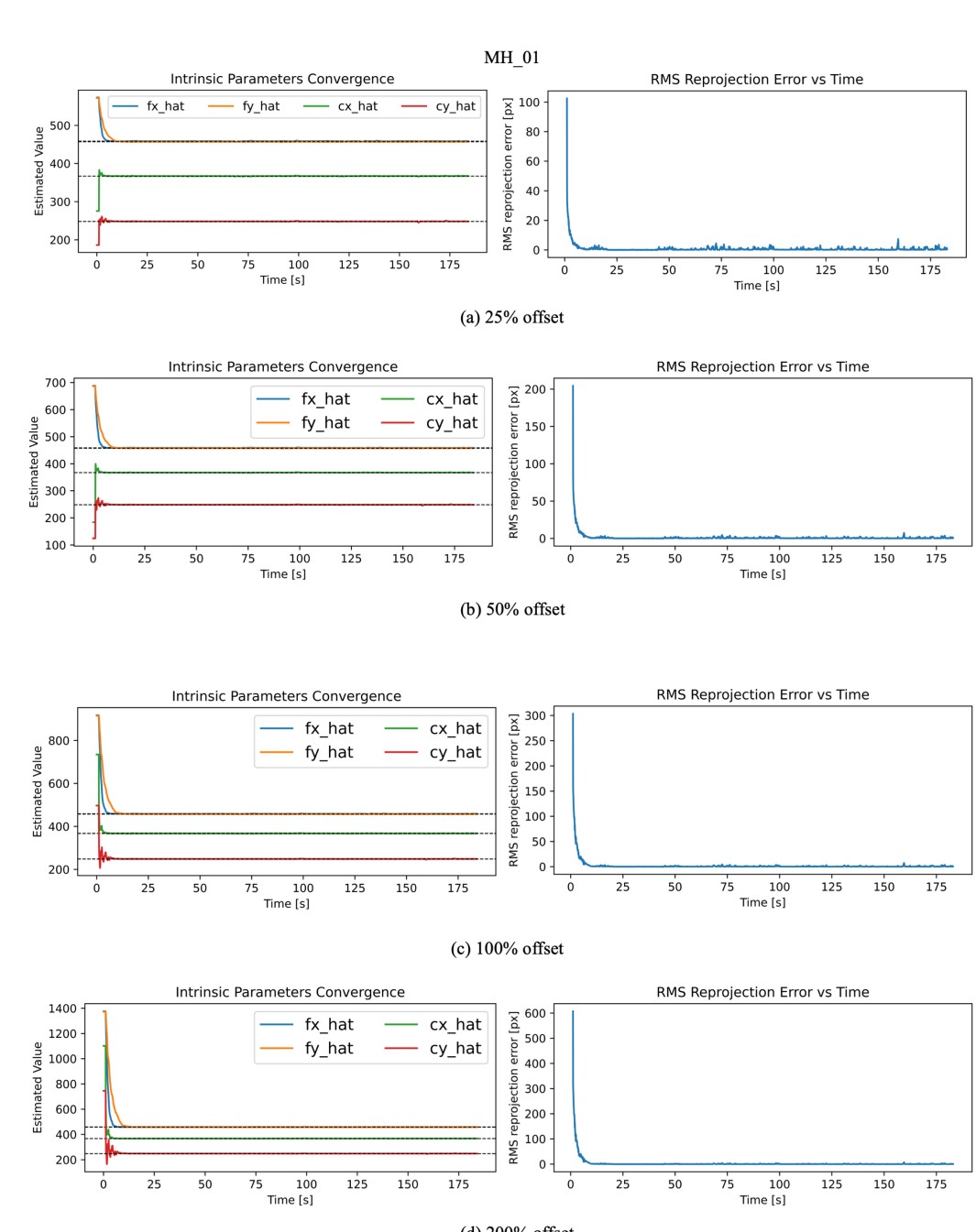

Figure 6: Convergence of Intrinsic parameters and RMS reprojection error for the MH_01 sequences under initial intrinsic-parameter offsets of (a) 25%, (b) 50%, (c) 100 and (d) 200%. In Intrinsic parameters convergence plots, each colored curve tracks error over time (in seconds), with the ground-truth intrinsics indicated by black dashed lines. Even with a 200% initial_offset, RACE drives error below 1% within 15s.

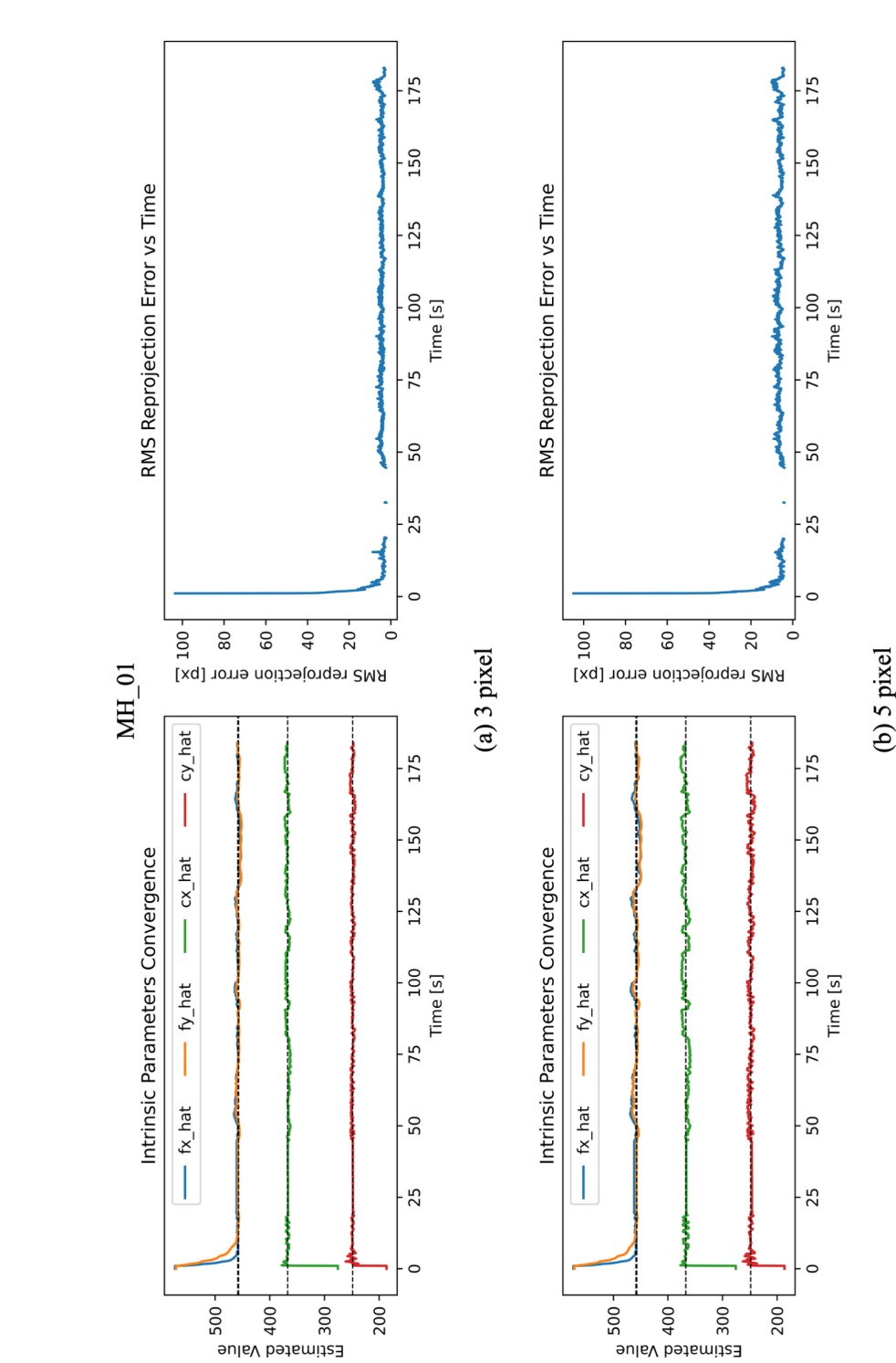

Figure 7: Measurement-Noise Robustness on EuRoC MAV (MH_01). We inject continuous zero-mean Gaussian noise with $\{3, 5\}$ px into feature coordinates. Despite continuous noise, RACE keeps RMS error fluctuations within bounded pixel values and never diverges. The increase in the ultimate error bound matches our Theorem 2, confirming quantitative robustness to measurement perturbations.

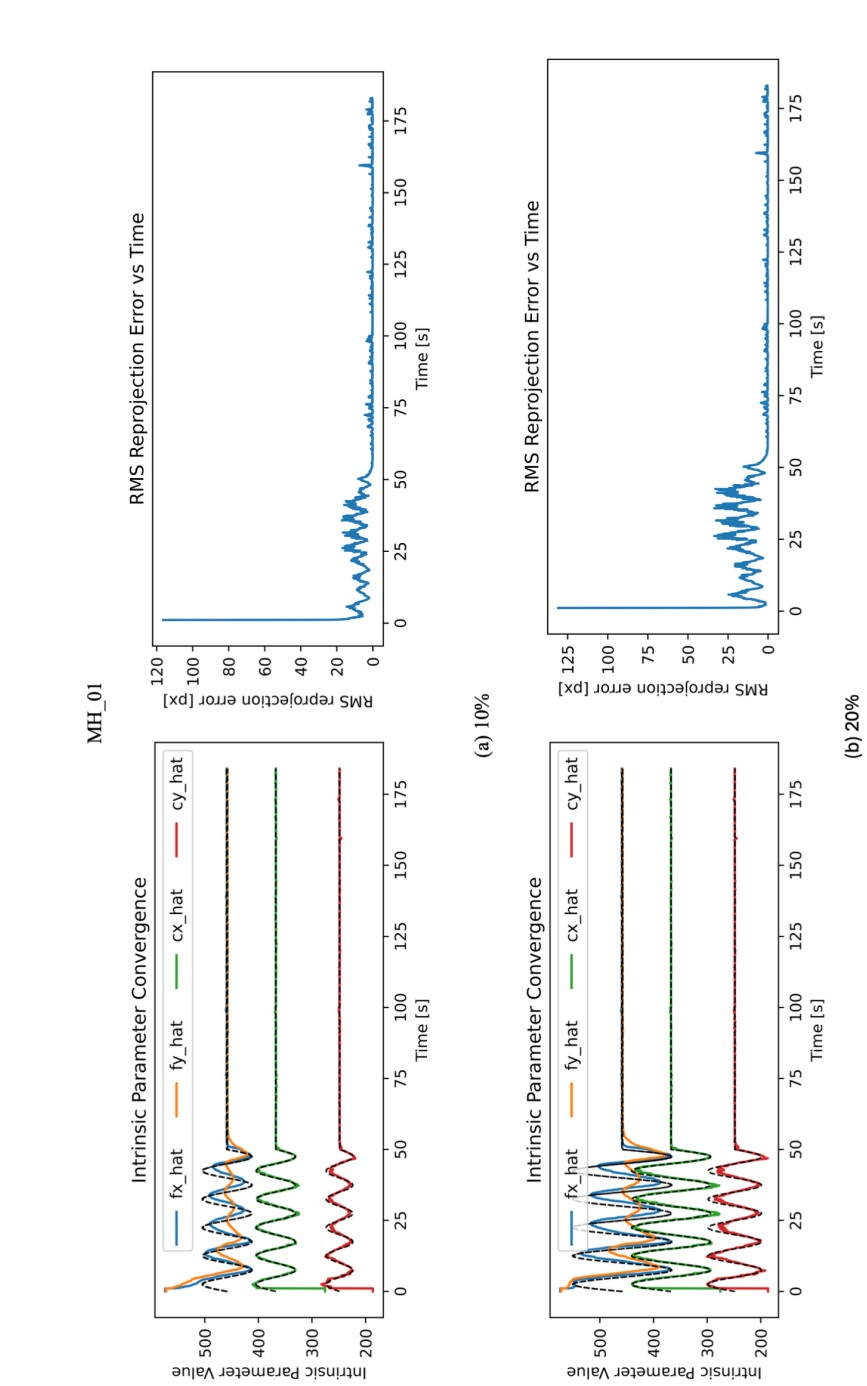

Figure 8: Thermal Drift Tracking on EuRoC MAV (MH_01). We apply a sinusoidal thermal pertur-
bation of amplitude $A_{\mathrm{therm}} \in \{10\,\%, 20\,\%\}$ and period $T_{\mathrm{therm}} = 10\,\mathrm{s}$ to all intrinsics, in addition to
default 25% offset initialization. RACE's estimated intrinsics (solid lines) continuously follow the
true drift, yielding sub-pixel RMS error (bottom subplot) throughout the 50 s experiment. **Note**: the
true intrinsic parameters vary according to the sinusoidal drift model C.7.

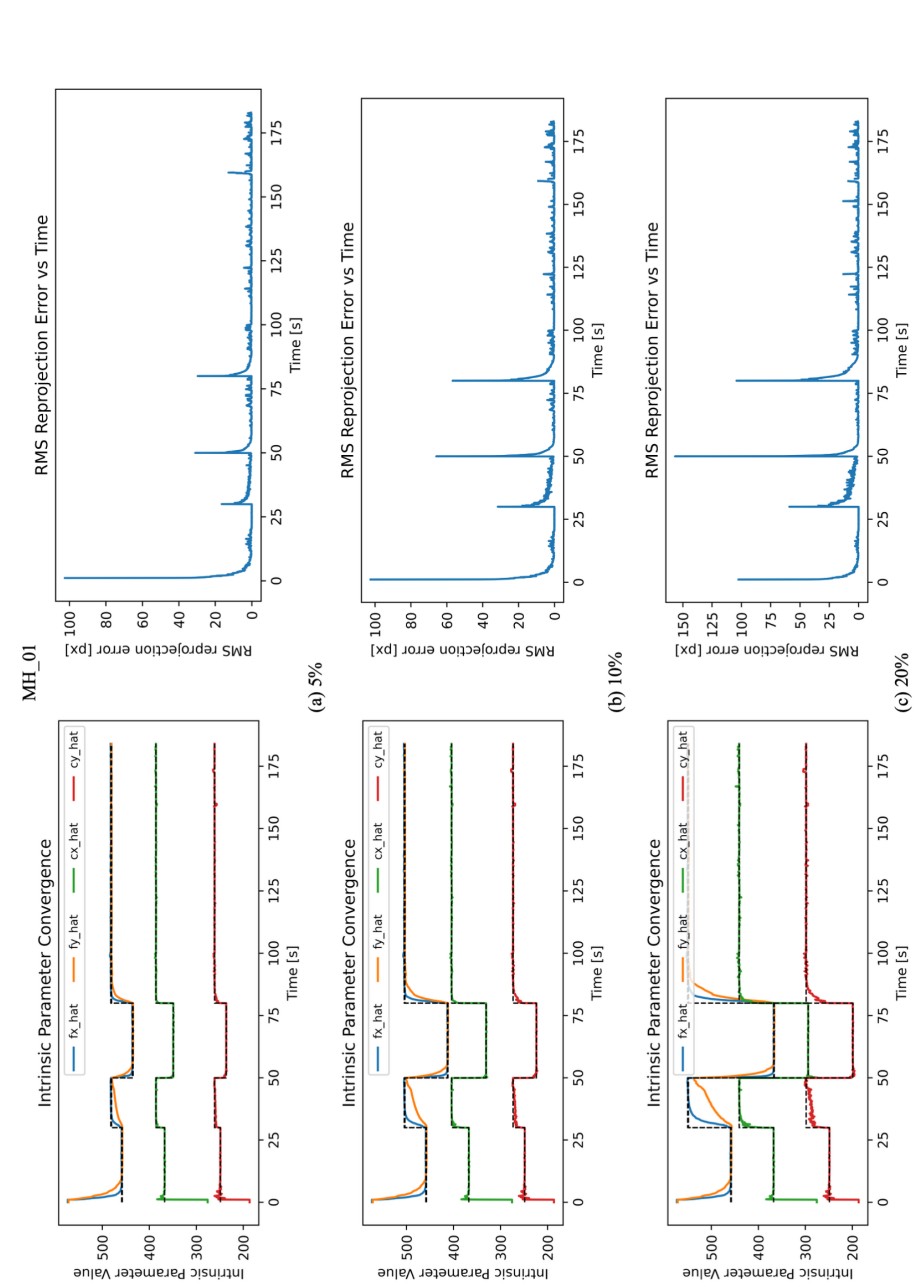

Figure 9: Plateau drift Recovery on EuRoC MAV (MH_01). At $t = \{30, 50, 80\}$ s we introduce step shifts of $\{5\,\%, 10\,\%, 20\,\%\}$ in all intrinsics. Each jump produces a sharp spike in ground truth intrinsic parameters and reprojection (error) error, but RACE reconverges to the new true values within 5-10 frames. This per-frame adaptability underscores its robustness to abrupt calibration shocks, a scenario where gradual methods fail catastrophically. **Note**: the true intrinsic parameters vary according to the Plateau drift model C.7.

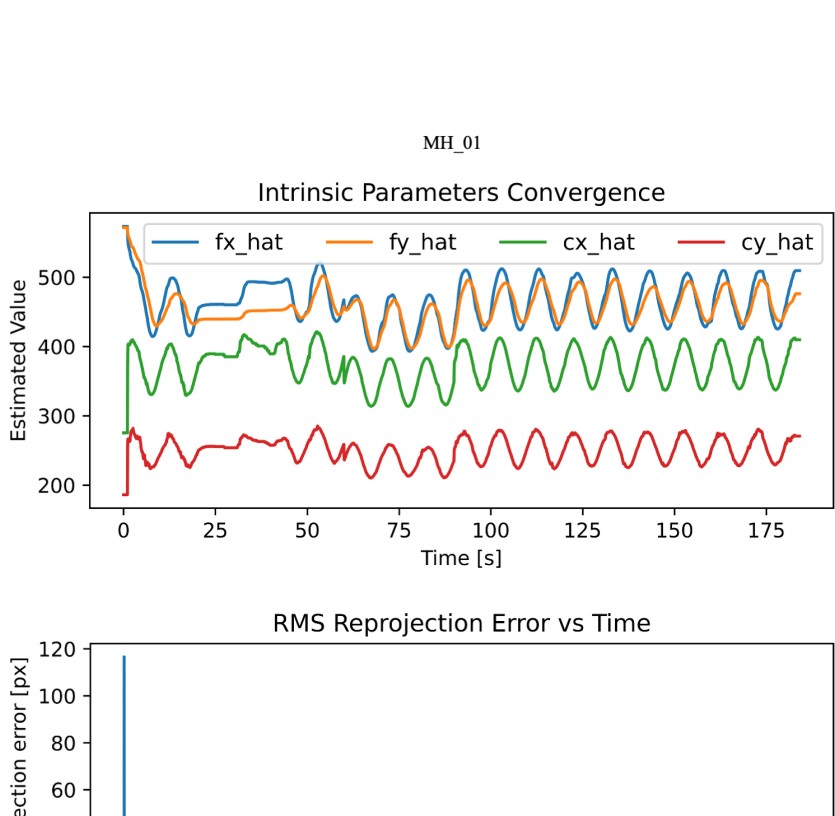

Figure 10: Combined Stress Test on EuRoC MAV (MH_01). Simultaneously applied disturbances include a 25% initial offset, 0.5 px Gaussian noise, 10% thermal drift (10 s period), and 5 % plateau drift at 30/60/90 s. RACE remains stable under this worst-case compound scenario, gracefully tracking the time-varying true intrinsic parameters and maintaining stable RMS error. **Note**: the true intrinsic parameters vary according to the drift added but are not shown in the plots for visual clarity.

