# OpenReview forum: "RACE: Real-Time Adaptive Camera-Intrinsics Estimation via Control Theory"
_ICLR.cc/2026/Conference — Submitted to ICLR 2026_

### Official Review · Reviewer_LMQH · 2025-10-27

**Soundness:** 3
**Presentation:** 3
**Contribution:** 3
**Rating:** 4
**Confidence:** 4

**Summary:**

The paper introduces a novel and highly original approach to online camera intrinsic calibration by framing it as an adaptive control problem. The core strength of the work lies in its rigorous theoretical foundation, providing formal stability guarantees (global asymptotic convergence and GUUB) under ideal conditions. The method is impressively efficient, achieving real-time performance on a single CPU core, and is training-free, offering strong potential for long-term autonomy.

**Strengths:**

RACE (Real-time Adaptive Camera-intrinsic Estimation) frames intrinsic calibration as an adaptive control problem. Treating intrinsics as dynamic states, RACE applies a lightweight Lyapunov-based update driven by reprojection errors. Under standard persistent excitation(PE), it proves global Lyapunov stability
of the error dynamics. Practically, RACE performs intrinsic calibration online, which is training-free, requires no bundle adjustment, and runs in real-time on a single CPU.

**Weaknesses:**

Despite its theoretical elegance, the paper suffers from several critical weaknesses that significantly undermine its claims of practicality and generalizability:

1.Overly Idealized Assumptions: The theoretical guarantees are predicated on the assumption of "known accurate poses," which creates a fundamental circular dependency in any real SLAM system. The analysis lacks any robustness guarantees under pose estimation noise, which is unavoidable in practice.

2.Inadequate Experimental Validation:
Unrealistic Perturbation Model: The use of a "global scaling" perturbation for initial offsets is a severe oversimplification. It avoids the core challenge of coupled parameter identifiability and likely overstates the method's robustness. The absence of tests with independent parameter perturbations is a major omission.
Unfair Benchmarking: The comparisons against state-of-the-art methods are not conducted on a level playing field. A direct comparison against classical online estimators (e.g., Recursive Least Squares, online Gauss-Newton) under the same simplified setting (known poses, same perturbations) is required to isolate the benefit of the proposed control law.
Lack of System Integration Clarity: The paper fails to clearly explain how the RACE module would be integrated into a full SLAM pipeline to break the circular dependency, leaving its practical implementation in doubt.

3.Incomplete Analysis of Limitations:
The extension to lens distortion relies on a heuristic "continual linearization" approach without providing stability proofs for the resulting nonlinear system.
The analysis of failure cases (e.g., on TartanAir) is superficial. The role and effectiveness of the proposed PE-gating mechanism are stated but not quantitatively demonstrated with ablation studies or diagnostic plots.

The paper presents a promising direction but in its current form, it reads more as a proof-of-concept under idealized conditions than a thoroughly validated practical solution. Addressing these points is crucial for transitioning the work from a theoretically interesting idea to a impactful contribution with clearly demonstrated real-world applicability.

**Questions:**

The following major revisions are essential for establishing the paper's credibility and practical value:
1.	Conduct a fair ablation study comparing RACE directly against Recursive Least Squares and an online Gauss-Newton optimizer under identical conditions (known poses, same perturbation models).
2.	Evaluate robustness using independent parameter perturbations to test performance in a more realistic and challenging scenario.
3.	Provide a clear description and analysis of how RACE can be integrated into a SLAM system without the "known pose" assumption, addressing the circular dependency problem.
4.	Include a thorough diagnostic analysis of failure modes (e.g., on TartanAir), providing quantitative evidence linking performance drops to PE condition violations and demonstrating the efficacy of the PE-gating mechanism.

---

> ### Author Response · Authors · 2025-11-22
>
> We thank the reviewer for their feedback and acknowledge the important points raised:
> 1. In the revised version (Appendix Sec. E), we include an ablation study where RACE, RLS, and Gauss Newton (GN) optimizer are evaluated under identical conditions (25% initial intrinsic perturbation and noise levels $\sigma \in {0, 5, 10}$).
> The results are:
> * RLS is highly sensitive to the forgetting factor $\lambda$. Small changes (e.g., $\lambda = 0.90$) lead to large variations in both reprojection and parameter error, and the method becomes unstable as noise increases.
> * Online GN is sensitive to step size. Larger steps cause overshoot or oscillation, moderate step sizes (e.g., $\alpha = 2.0$) lead to divergence with intrinsic parameter errors exceeding 200–300 pixels, and GN remains sensitive to noise even when tuned.
> * RACE is stable across all settings. It achieves consistently low final RMS and parameter error across noise levels, outperforming RLS and GN and remaining competitive with carefully tuned variants.
>
> 2. To evaluate robustness under independent parameter perturbations, we conducted 10 trials of RACE on the EuRoC. In each trial, the initial intrinsic parameters were independently perturbed by sampling a random offset within ${-30%, +30%}$ of each ground-truth parameter, producing different and uncorrelated initial intrinsics across trials.
> Across all trials, RACE consistently converged to the correct intrinsics. We report the mean $\pm$ standard deviation across trials for the key metrics, including:
> (i) the minimum reprojection error,
> (ii) the time required for the parameter error to fall below 5% and 1% of its initial value,
> (iii) the final intrinsic parameter values, and
> (iv) the final intrinsic-parameter error (L2 norm).
> These results show that RACE remains stable and reliable under independent perturbations of each intrinsic parameter, demonstrating strong robustness to realistic variations in initialization. These results are included in the Appendix C.3.
>
> 3. As explicitly stated in the paper, full SLAM integration remains future work. However, we outline our research direction and how RACE can complement visual SLAM and odometry. Our first goal is to establish RACE as a theoretically grounded online calibration, and the next is to relax the “known-pose” assumption.
>
>    Building on Appendix C.5, we show that the “known pose” assumption can be relaxed in several practical ways. One approach is to estimate the camera pose analytically using a calibration target board (with known 2D-3D correspondences[1]), and then apply RACE to refine the intrinsics. To evaluate this idea, we developed a simulation setup that mirrors this real world scenario. We first compute an initial extrinsic estimate from a single image of target board and then run RACE. This procedure enables full intrinsic calibration from just one frame, the resulting accuracy is competitive with standard offline calibration with just one frame.  Additional details are included in Appendix G.
>
>    After relaxing the pose requirement, the remaining assumption is access to reliable 2D-3D correspondences. We envision addressing this using a lightweight learning based model that predicts correspondences directly from unconstrained images. Overall, this show the pathway of fusing the control theory guarantee with learning based generalization for online estimation of pose and camera calibration. Importantly, we have also shown robustness against the pose estimation error in Appendix C.4.
>
> [1] Zhang Z. A flexible new technique for camera calibration. IEEE Transactions on pattern analysis and machine intelligence. 2002 Aug 6;22(11):1330-4.
>
>
> 4. In the initial TartanAir evaluation we used a very small PE-gating threshold $PE \ge 0.1$, which allowed degraded updates Supplementary Sec. 1.3.1. After re-running TartanAir sequences (MH000) with $PE \ge 5$, we observed a clear reduction in both intrinsic parameters and reprojection error spikes, demonstrating the effectiveness of PE-gating.
>
>    However, PE-gating alone does not guarantee smooth convergence of $\hat{\theta} \to \theta^*$ because TartanAir exhibits  intervals of low illumination leading to severely ill-conditioned $\Phi_t$, all of which we identify as primary failure modes  Appendix C.6.
>
>    To further isolate the cause, and consistent with Appendix C.5, we selected a single TartanAir MH000 frame satisfying: (i) $\ge 200$ valid 2D-3D correspondences, and (ii) $PE \ge 0.9$. Running RACE for 25K static iterations with this frame (fixed 2D-3D correspondence, known pose) and observe produced smooth convergence of both intrinsic and reprojection error. This confirms that when the regressor is well-conditioned and even with lower PE, RACE exhibits stable convergence, suggesting that failures on TartanAir arise from low illumination making regressor $\Phi_t$ ill-conditioned. Additional quantitative diagnostics correlating errors with PE values are provided in Appendix C.6.

---

> > ### Author Response · Authors · 2025-11-29
> >
> > Thank you for the thoughtful and constructive feedback. We have carefully addressed all of your concerns in the author response and updated the text and experiments accordingly. We hope that our clarifications resolve the issues you raised.
> >
> > We would be happy to provide any additional clarifications or answer further questions. Otherwise, in light of the updated explanations and results, please consider revisiting your evaluation score.

---

### Official Review · Reviewer_47TU · 2025-10-31

**Soundness:** 4
**Presentation:** 3
**Contribution:** 4
**Rating:** 8
**Confidence:** 4

**Summary:**

1. The paper proposes that intrinsics will drift in real-world operation, due to heating, zoom events, mechanical shocks.

2. This work presents RACE, a control-theory-based online camera intrinsic estimator that runs in real time on a single CPU (no GPU/pre-training/batch processing). It provides theoretical guarantees (global stability, convergence) and outperforms baselines (e.g., COLMAP, DroidCalib) on EuRoC/TUM RGB-D with sub-pixel reprojection error, addressing real-world calibration pain points.

3.  Grounded in control theory, RACE treats intrinsic parameters as dynamic states and employs a lightweight Lyapunov-based update law. RACE requires no training data, bundle adjustment, or retraining, and provides the first theoretical bridge between adaptive control and online learning for camera models. RACE is a new class of theoretically grounded continual learners for camera intrinsics, enabling robust long-term perception in embodied agents.

**Strengths:**

1. Novel insight: camera intrinsics will drift in real-world operation, due to heating, zoom events, mechanical shocks in real-world application.
2. Real-time and online calibration
3. Less projection errors by allowing for online update of camera intrinsics

**Weaknesses:**

1. Require precise camera extrinsics such as Rotation and Translation matrix.
2. Rely on persistent excitation to guarantee convergence and stability.

**Questions:**

1. Suppose that the colmap could update its predicted camera intrinsics, could RACE still surpass colmap in precision?

---

> ### Author Response · Authors · 2025-11-22
>
> Even if COLMAP were modified to continuously update its estimated intrinsics, RACE would still offer advantages. COLMAP’s bundle adjustment (BA) optimizes intrinsics jointly with poses and structure, but its accuracy is fundamentally limited by (i) delayed updates that require full batch windows, (ii) sensitivity to initialization, and (iii) degraded performance under time varying or drifting intrinsics, for which standard BA has no explicit stability guarantees.
>
> RACE, in contrast, performs per-frame intrinsic updates with adaptive law that allows RACE to correct intrinsic drift immediately when new information appears, without requiring a batch optimization and with theoretical convergence guarantees. As a result, even if COLMAP were allowed to update intrinsics, RACE can still have some edge in precision, especially in scenarios involving gradual drifting intrinsics and where frequent re-linearization is expensive.

---

> > ### Author Response · Authors · 2025-11-29
> >
> > Thank you for highlighting the strengths and novelty of our paper, particularly the ability of our method to perform real-time online calibration, its robustness to uncertain drifts and shocks, and its improved accuracy over existing approaches. We have addressed your question regarding COLMAP in our response. If any further clarification would be helpful, we would be happy to provide it.

---

### Official Review · Reviewer_qHGu · 2025-11-01

**Soundness:** 3
**Presentation:** 3
**Contribution:** 3
**Rating:** 4
**Confidence:** 2

**Summary:**

This paper introduces RACE, a new method for estimating camera intrinsic parameters (like focal length and principal point) online and in real-time. The core idea is to frame this as an adaptive control problem. The authors use a Lyapunov-based update law to prove that their estimator is stable and that the intrinsic parameter error will asymptotically converge to zero, provided the camera's motion and scene are sufficiently varied (a condition known as persistent excitation).

The method does not require any pre-training or large datasets. The paper's experiments on benchmarks (EuRoC, TUM) show that RACE is fast (running on a CPU), robust to large initial errors, and achieves accuracy that matches or surpasses other state-of-the-art learning-based methods.

**Strengths:**

1. Theoretical Novelty: The primary strength is the use of adaptive control theory to tackle this problem. Applying a Lyapunov-based analysis provides provable guarantees of stability and convergence, which is a significant advantage over many end-to-end deep learning methods that may lack such guarantees.

2. Efficiency and Accessibility: The method is very lightweight. It runs in real-time on a single CPU core, requires no GPU, and is training-free. This makes it highly practical from a computational standpoint.

3. Strong Experimental Results (Given Assumptions): The empirical results are impressive. The method demonstrates high accuracy on the EuRoC and TUM datasets, and the ablation studies show that it is robust to significant noise and very large initial parameter errors (e.g., 100-200% offsets), which is a strong validation of its stability.

**Weaknesses:**

1. Strong Assumption of Known Poses: The most significant weakness is the assumption that the algorithm has access to accurate camera poses and 2D-3D correspondences. The authors state this in Section 3.2 and the limitations. This assumption seems to create a 'chicken-and-egg' problem for the scenarios the paper motivates, like autonomous driving or robotics. In a real-world setting, a drift in camera intrinsics would almost certainly degrade the performance of the pose estimation system (e.g., SLAM or localization). One cannot assume a perfect pose to fix the intrinsics, because the imperfect intrinsics are needed to find the pose. This assumption, while also made by some other works, severely limits the practical applicability of the method as a standalone solution.
2. Reliance on Persistent Excitation (PE): The method's convergence guarantee depends on the PE condition, meaning the camera must be moving in a way that provides rich visual information. The paper's own results on the TartanAir dataset show that performance degrades in challenging segments (fog, low light) where this condition is likely not met. While the authors propose 'gating' (pausing) the update, this remains a practical limitation for real-world use where long, non-informative sequences can occur (e.g., driving on a straight highway).

**Questions:**

1. Given the reliance on known camera poses, could the authors elaborate on the intended practical deployment scenario? Is this method intended to be integrated into a larger SLAM or visual odometry system?

2. Following up on the first question, Appendix E shows a test with ORB-SLAM3. Can you clarify the setup? Was the RACE algorithm still fed ground-truth poses to estimate intrinsics, which were then passed to ORB-SLAM3? Or was there a joint estimation where RACE used the (potentially incorrect) estimated poses from ORB-SLAM3? How would the stability guarantees be affected if the input poses are noisy or drifting?

3. The paper proposes 'gating' the update when the PE condition is weak. Was this technique used in the TartanAir experiments? How well does the system perform if it encounters a long period of weak PE and then must re-converge when informative motion resumes?

---

> ### Author Response · Authors · 2025-11-22
>
> We thank the reviewer for their feedback and acknowledge the important points raised. Below, we address the key concerns raised:
> 1. The current formulation assumes access to known poses and accurate 2D-3D correspondences, was made intentionally so that we could isolate and analyze the intrinsic-parameter convergence and stability properties of RACE. Our first goal is to establish RACE as a theoretically grounded online camera calibration, and next is relaxing known pose assumption.
>
>    Building on Appendix C.5 (intrinsic estimation from a single frame), we show that the “known pose” assumption can be relaxed in several practical ways. One approach is to estimate the camera pose analytically using a calibration target (known 2D-3D correspondences [1]), and then apply RACE to refine the intrinsics. To evaluate this idea, we developed a simulation setup that mirrors this real world scenario. We first compute an initial extrinsic estimate from a single image of target board and then run RACE. This procedure enables full intrinsic calibration from just one frame, and as summarized in the Table 1, the resulting accuracy is competitive with standard offline calibration. Additional details are included in Appendix Sec G.
>
>    After relaxing the pose requirement, the remaining assumption for real world deployment is access to reliable 2D-3D correspondences. We envision addressing this through lightweight learned modules that predict such correspondences directly from images. This show the pathway of fusing the control theory guarantee with learning based generalization for online estimation of pose and camera calibration. In the future work will be explore the integrate of RACE with visual odometry and modern SLAM systems.
>
>    [1] Zhang Z. A flexible new technique for camera calibration. IEEE Transactions on pattern analysis and machine intelligence. 2002 Aug 6;22(11):1330-4.
>
>    **Table 1** True intrinsics: (800,800,320,240). Single-image Zhang calibration shows large deviations in focal lengths. RACE with the true pose converges exactly to the correct intrinsics. RACE using only the pose estimated from Zhang's single-image solution. A standard 15-image Zhang calibration with nonlinear optimization.
> |Method|EstimatedIntrinsics(fx,fy,cx,cy)|FinalRMS(px)|‖θ_est−θ_true‖₂|
> |---|---|---|---|
> |Zhang(1img)|(767.54,763.81,320.00,240.00)|1.4|47.55|
> |RACE(TruePose)|(800.00,799.999999,320.00,240.00)|0.00|0.001|
> |RACE(EstPose)|(801.3,796.8,319.94,240.4)|0.14|3.46|
> |Zhang(15img)|(800.322,800.337,320.239,240.373)|0.495|0.661|
>
>
> 2 Here, ORB-SLAM3 is started with a deliberately mis-scaled intrinsics (25\% bias). The SLAM backend (standard ORB-SLAM3) receives exactly the same type of inputs as the baseline, ie. a monocular image stream plus a single YAML calibration. The only difference is that we pre-warp each raw frame with the per-frame intrinsic log via RACE before feeding it to SLAM.
>
>    In Appendix F, we compared ORB-SLAM3 (baseline), ORB-SLAM3 with biased intrinsic parameters and ORB-SLAM3 with RACE. Compared to the ORB-SLAM3 with biased intrinsic parameters, ORB-SLAM3 with RACE reduces ATE RMSE significantly and brings the error to within a few centimeters of the fully calibrated oracle run. Thus, the proposed calibration method works even when the SLAM system begins with incorrect intrinsic estimates.
>
>    For stability guarantees w.r.t noisy input poses, our manuscript contains ablation studies in  Appendix C.4. We perturbed every 3D landmark in every frame of the sequence with i.i.d. Gaussian noise $N(0,\sigma^2 I_3)$, where $\sigma$ in {1,3,5,10} cm, and added 25\% initial bias in parameters. Increased noise amplified reprojection error, causing larger adaptation steps and faster initial convergence in high-noise scenarios. However, noise didn’t improve accuracy, where final convergence precision degraded with higher noise levels.
>
>    For further clarification, we have expanded on this in Appendix F. This setup is designed to isolate the benefit of correcting intrinsics on SLAM performance, serving as a controlled evaluation before pursuing full SLAM-integrated calibration, which will be conducted in our future work.
>
> 3. Yes. In the initial TartanAir evaluation we used a very small PE-gating threshold $PE \ge 0.1$, which allowed degraded updates. After re-running TartanAir sequence MH000 with a stricter gate $PE \ge 5$, we observed a clear reduction in both intrinsic parameter error and reprojection error spikes Appendix C.6. It should be noted here that, PE-gating freezes the updates but does not enforce smooth convergence, since TartanAir contains highly intermittent excitation, low illumination, and ill-conditioned regressor (Supplementary  1.3.1), resulting in degraded performance.
>
>     As shown in Appendix C.6 , we did freeze the update from 50th to 500th frame, during this frozen interval the intrinsics remain stable without drifting, and once informative motion returns, the true parameters are recovered.

---

> > ### Author Response · Authors · 2025-11-29
> >
> > Thank you for the thoughtful and constructive feedback. We have carefully addressed all of your concerns in the author response and updated the text and experiments accordingly. We hope that our clarifications resolve the issues you raised.
> >
> > We would be happy to provide any additional clarifications or answer further questions. Otherwise, in light of the updated explanations and results, please consider revisiting your evaluation score.

---

### Official Review · Reviewer_65E9 · 2025-11-02

**Soundness:** 1
**Presentation:** 2
**Contribution:** 1
**Rating:** 2
**Confidence:** 4

**Summary:**

In this paper, the authors propose a real-time adaptive camera-intrinsic estimation (RACE) method.

Different from previous offline approaches, self-calibration methods with global bundle adjustment (e.g. colmap), and pretrained models, the proposed method updates the intrinsic parameters adaptively based on control theory with a strong assumption that correct 2D-3D correspondences and camera poses are known.

Experiments on public datasets including EuRoc, TUM RGB-D, and TartanAir datasets demonstrate the proposed approach gives smaller reprojection errors than previous methods such as Colmap, DroidCalib which don’t have any assumptions.

**Strengths:**

1.	Good motivation. As mentioned in  the paper, the camera intrinsics parameters change with time (maybe not too much) and require careful calibration before being use for data collection. Previous works including Zhang’s approach, global BA, and end-to-end models do have their disadvantages. Therefore, a simple, effective, and fast online calibration approach is very useful.

2.	Experiments on three public datasets are good. Maybe one or two outdoor datasets would be better.

3.	The metric of reprojection error makes sense especially with the strong assumption that correct 2D-3D correspondences and camera poses are known.

**Weaknesses:**

1.	Very strong and impractical assumption. The proposed method assumes that correct 2D-3D correspondences and camera poses are known. This is a extremely strong assumption and is impractical in real applications. How can we get correct 2D-3D correspondences and camera poses without knowing the correct intrinsic parameters?

2.	If correct 2D-3D correspondences and camera poses are given as assumed in the paper, the easiest way to obtain the intrinsic parameters is solving a least square problem (LSP). What is difference between the proposed method and the least square problem? Besides, the LSP should be one baseline for comparison in the experiments.

3.	The methods chosen for comparison including colmap, DroidCalib, etc  never have assumptions of 2D-3D correspondences or camera poses. The comparison between the proposed approach with these methods is not fair.

4.	The reprojection error with ground-truth camera intrinsic parameters should be included in experiments as reference as their results should be the upbound.

**Questions:**

Overall, I don’t think this paper should be accepted. Please see the weaknesses section for more details.

---

> ### Author Response · Authors · 2025-11-22
>
> We thank the reviewer for their feedback and acknowledge the important points raised. Below, we address the key concerns raised:
>
> 1. The current formulation assumes access to known poses and accurate 2D-3D correspondences, was made intentionally so that we could isolate and analyze the intrinsic-parameter convergence and stability properties of RACE. Our first goal is to establish RACE as a theoretically grounded online camera calibration, and next is relaxing known pose assumption.
>
>    Building on Appendix C.5 (intrinsic estimation from a single frame), we show that the “known pose” assumption can be relaxed in several practical ways. One approach is to estimate the camera pose analytically using a calibration target (known 2D-3D correspondences [1]), and then apply RACE to refine the intrinsics. To evaluate this idea, we developed a simulation setup that mirrors this real world scenario. We first compute an initial extrinsic estimate from a single image of target board and then run RACE. This procedure enables full intrinsic calibration from just one frame, and as summarized in the Table 1, the resulting accuracy is competitive with standard offline calibration. Additional details are included in Appendix Sec G.
>
>    After relaxing the pose requirement, the remaining assumption for real world deployment is access to reliable 2D-3D correspondences. We envision addressing this through lightweight learned modules that predict such correspondences directly from images. This show the pathway of fusing the control theory guarantee with learning based generalization for online estimation of pose and camera calibration. In the future work will be explore the integrate of RACE with visual odometry and modern SLAM systems.
>
> [1] Zhang Z. A flexible new technique for camera calibration. IEEE Transactions on pattern analysis and machine intelligence. 2002 Aug 6;22(11):1330-4.
>
> **Table 1** True intrinsics: (800,800,320,240). Single-image Zhang calibration shows large deviations in focal lengths. RACE with the true pose converges exactly to the correct intrinsics. RACE using only the pose estimated from Zhang's single-image solution. A standard 15-image Zhang calibration with nonlinear optimization.
> |Method|EstimatedIntrinsics(fx,fy,cx,cy)|FinalRMS(px)|‖θ_est−θ_true‖₂|
> |---|---|---|---|
> |Zhang(1img)|(767.54,763.81,320.00,240.00)|1.4|47.55|
> |RACE(TruePose)|(800.00,799.999999,320.00,240.00)|0.00|0.001|
> |RACE(EstPose)|(801.3,796.8,319.94,240.4)|0.14|3.46|
> |Zhang(15img)|(800.322,800.337,320.239,240.373)|0.495|0.661|
>
> 2. LS is often considered for estimation because it can attenuate noise by solving a batch regression over many measurements. The Least Squares (LS) is not equivalent to the RACE update for three key reasons:
>
> * RACE is recursive and causal: RACE updates intrinsics per-frame in time without ever solving a batch problem. LS requires accumulating a window and re-optimizing each time.
>
> *  RACE provides guaranteed stability and convergence: Our update law is derived from a Lyapunov function with formal guarantees (Theorems 1-2).
>
> *  RACE explicitly tracks per-frame time varying intrinsics with proven boundedness, where LS assumes a static parameter.
>
> These differences are clearly demonstrated in the time-varying intrinsics experiment Table 2. LS diverges rapidly and shows no ability to track continuously changing parameters. In contrast, RACE continuously minimizes the reprojection error at every frame and reliably tracks the intrinsic parameters as they evolve over time, maintaining precise alignment with the drifting ground truth. Additional details and figures are provided in Appendix Sec. E.
>
> **Table 2**:Performance comparison between RACE and LS under time drift in intrinsics. RACE tracks drifting intrinsics; LS fails.
> |Method|FinalRMS[px]|
> |---|---|
> |RACE|6.6|
> |LS|47.2|
>
> 3. Our intention was not to claim that RACE replaces these systems. Instead, the comparison evaluates intrinsic precision when ground truth geometry is available. The motivation is: COLMAP and DROIDCalib are typically used to “fix” calibration for downstream task. RACE aims to match or exceed the intrinsic accuracy of these methods, even though RACE operates online, handle noise, parameter drift and require no training data.
>
> 4. We thank the reviewer for the suggestion. While a single ground truth reprojection error across entire sequences cannot be provided due to per-frame noise and other uncertainties, but we will include the dataset provided ground truth intrinsic parameters and compare our estimated intrinsics against them in the camera ready version.

---

> > ### Comment · Reviewer_65E9 · 2025-11-25
> > **Assumption of known camera poses and 2D-3D correspondences**
> >
> > Thanks for the update.
> >
> > Could you please provide an implementation that does not require accurate camera poses and 2D-3D correspondences? In practice, neither of them is provided (e.g., VO/SLAM systems).

---

> > > ### Author Response · Authors · 2025-11-28
> > >
> > > Thank you for raising this important question about removing the dependence on accurate poses and 2D–3D correspondences:
> > >
> > > As shown in Appendix G, a single well conditioned frame with known 2D–3D correspondences is sufficient for estimating both intrinsic and extrinsic (pose) parameters. In our simulation experiments (Appendix G, Table 17), we demonstrate that the extrinsic (pose) parameters can be computed analytically from a calibration target, after which RACE operates without any external pose source.
> > >
> > > Traditional target based calibration pipelines require 15-25 images captured from multiple viewpoints, followed by an offline batch optimization. This process is time consuming, interruptive, and impractical to repeat during ongoing vision based operation. In contrast, our method enables online calibration from a single target board, making the procedure simple, lightweight, and suitable for real world deployment. This demonstrates the first step toward practical, in the wild application of RACE.
> > >
> > > To date, we have shown that RACE can reliably estimate both extrinsic (pose) and intrinsic parameters online while benefiting from the global convergence and stability guarantees of adaptive control theory. The only remaining assumption is access to 2D-3D correspondences (i.e., a known calibration target).
> > >
> > > In future work (Appendix G), we plan to relax the 2D–3D correspondence requirement, enabling RACE to jointly estimate extrinsics (pose) and intrinsics online in unstructured environments. Appendix G also outlines a potential pathway for extending RACE to operate without explicit calibration targets. However, developing a full VO/SLAM system that RACE complements or replaces is yet to be explored and is left for future work.

---

> > > > ### Author Response · Authors · 2025-11-29
> > > >
> > > > We appreciate the thoughtful and constructive feedback. We have carefully addressed all of your concerns in the author response and updated the text and experiments accordingly. We hope that our clarifications resolve the issues you raised.
> > > >
> > > > We would be happy to provide any additional clarifications or answer further questions. Otherwise, in light of the updated explanations and results, please consider revisiting your evaluation score.

---

### Author Response · Authors · 2025-12-02
**Summary Comment**

We thank all reviewers for their thoughtful and detailed evaluations.

- Reviewer 47TU strongly emphasizes the technical contributions: **theoretical guarantees** (global stability and convergence), training-free operation, real-time online calibration on **a single CPU**, robustness to drift, and **improved accuracy over baselines** (COLMAP/DroidCalib).

 - Reviewers qHGu and LMQH highlight the **theoretical novelty**, **strong experimental results**, and practical promise of **framing camera calibration as an adaptive control problem**.

 - Reviewer 65E9 appreciated the **motivation and evaluation metric** while raising questions regarding assumptions and baselines. We addressed these through new experiments, including a **real-world deployment scenario and additional baselines**, which the reviewer acknowledged.

## Summary of Rebuttal
---
### (A) **Known Pose and 2D–3D Correspondence Assumption** (Reviewer 65E9, qHGu, LMQH)
We clarified that the initial “known pose” assumption was intentional to isolate and rigorously analyze RACE’s convergence and stability properties. **Appendix G** demonstrates how this can be **relaxed in practice** : using a single well conditioned frame, we estimate pose analytically from a standard calibration target (2D–3D correspondences) and then run **RACE to refine intrinsics**. This single-frame procedure achieves **accuracy competitive with offline calibration** and provides **a practical deployment path without assuming known poses**. Relaxing the remaining need for 2D–3D correspondences is future work, and we outline a clear path using lightweight learned correspondence predictors to combine control-theoretic guarantees with learning-based perception.

### (B) **Additional Baselines (LS/RLS/GN)** (Reviewer LMQH, 65E9):
In Appendix E, we added a dedicated ablation comparing RACE, LS, RLS, and GN under identical conditions: known poses, identical noise levels, and the same intrinsic perturbations. These experiments show that RLS and GN are highly sensitive to hyperparameters and noise, whereas **RACE remains stable and achieves consistently low reprojection and parameter error**. We also clarify why LS is not equivalent to RACE: LS is batch, static, and lacks stability guarantees for drifting parameters, while **RACE is recursive, causal, and designed to track time-varying intrinsics with formal guarantees**.

### (C) **Robustness under Independent Perturbations and Noise** (Reviewer LMQH):
We added experiments where each intrinsic parameter is **independently perturbed by a random offset in {–30%, +30%}** across 10 trials. The results show that RACE consistently converges to the correct intrinsics under independent and realistic perturbations (Appendix C.3).

### (D) **PE, PE-gating, and Failure Modes** (Reviewers qHGu, LMQH):
We clarify that PE-gating was indeed used in the TartanAir experiments and added ablations where we tighten the PE threshold, demonstrating clear **reductions in intrinsic and reprojection error spikes** (Appendix C.6, Supplementary 1.3.1). We show that during long weak-PE segments, the parameters remain stable, and **RACE quickly reconverges the intrinsics once PE rises above the threshold**. We further demonstrate convergence on a single well-conditioned frame iterated 25k times, confirming that when excitation and conditioning are adequate, RACE exhibits smooth, stable convergence.

---
In **conclusion**, we believe the submission presents a **coherent and well substantiated contribution**: a provably stable, real time, training free, CPU-based intrinsic calibration method, supported by **formal theoretical analysis** and **a strengthened experimental** section that directly addresses the reviewers’ main concerns (LS/RLS/GN baselines, independent perturbations, PE-gating, SLAM interaction, and failure-mode diagnostics). Although reviewer engagement in the discussion phase was limited, we have carefully addressed each concern through new experiments and theoretical discussion. We trust that the Area Chair will take this context into account. **Taken together, the theoretical guarantees, practical efficiency, and expanded empirical evidence make RACE a strong fit for ICLR, and we respectfully hope the Area Chair will view it as a valuable and timely contribution to robust perception**.

---

### Meta-Review · Area_Chair_2B6B · 2026-01-11

**Summary:**

While the reviewers appreciate the novel idea of framing camera calibration as an adaptive control problem and the theoretical guarantees, they raised concerns about the strong/impractical assumptions, the lack of comparisons/analyses, and the unconvincing evaluations. The authors’ rebuttal addressed some of them but failed to assuage the major concerns, in particular, the strong assumptions about the availability of known camera poses and 2D–3D correspondences. In the rebuttal, the authors (Appendix G) demonstrated that the pose assumption can be relaxed using a single well-conditioned frame, but they left the assumption of 2D–3D correspondences for future work. Considering all the reviews and the rebuttal, AC sides with the negative reviewers (65E9, qHGu, LMQH) and feels that, despite its promising potential, the paper, in its current form, reads as an immature work under idealized conditions rather than a validated practical solution. AC thus recommends rejecting the current version and encourages the authors to improve their work by thoroughly addressing the reviewers’ points and clearly demonstrating real-world applicability.

**Reviewer Concerns:**

The main concerns were about the strong/impractical assumptions, the lack of comparisons/analyses, and the unconvincing evaluations.
The authors’ rebuttal addressed some of them but failed to assuage the major concerns, in particular, the strong assumptions about the availability of known camera poses and 2D–3D correspondences. The issue of real-world applicability remains still outstanding.

**Reviewer Scores:**

Reviewer 65E9 retained the original rating of 2 after discussion.
Reviewer qHGu would have retained the original rating of 4.
Reviewer 47TU  would have retained the original rating of 8 or lowered it to 6.
Reviewer LMQH would have retained the original rating of 4.

---

### Decision · Program_Chairs · 2026-01-26

Reject